# Effects of seven days' fasting on physical performance and metabolic adaptation during exercise in humans

Kristoffer J. Kolnes[1,2], Emelie T. F. Nilsen[1,12], Steffen Brufladt[1,12], Allison M. Meadows[3,4], Per B. Jeppesen[5], Øyvind Skattebo[1], Egil I. Johansen[1], Jesper B. Birk[6], Kurt Højlund[2], Janne Hingst[6], Bjørn S. Skålhegg[7], Rasmus Kjøbsted[6], Julian L. Griffin[3,8], Anders J. Kolnes[9,10], Stephen O'Rahilly[11], Jørgen F. P. Wojtaszewski[6] & Jørgen Jensen[1]✉

Humans have, throughout history, faced periods of starvation necessitating increased physical effort to gather food. To explore adaptations in muscle function, 13 participants (7 males and 6 females) fasted for seven days. They lost $4.6 \pm 0.3$ kg lean and $1.4 \pm 0.1$ kg fat mass. Maximal isometric and isokinetic strength remained unchanged, while peak oxygen uptake decreased by 13%. Muscle glycogen was halved, while expression of electron transport chain proteins was unchanged. Pyruvate dehydrogenase kinase 4 (PDK4) expression increased 13-fold, accompanied by inhibitory pyruvate dehydrogenase phosphorylation, reduced carbohydrate oxidation and decreased exercise endurance capacity. Fasting had no impact on 5' AMP-activated protein kinase (AMPK) activity, challenging its proposed role in muscle protein degradation. The participants maintained muscle strength and oxidative enzymes in skeletal muscle during fasting but carbohydrate oxidation and high-intensity endurance capacity were reduced.

Throughout history, humans have faced periods of nutritional shortage, necessitating increased physical effort to gather new sources of food. Humans are well-adapted to tolerate periods without food and most individuals have sufficient fat stores to survive several weeks[1], but six days' fasting decreases lean mass substantially[2], which may impair physical capability. During fasting, the degradation of protein is the main source of amino acids for gluconeogenesis[3]. Whether this protein degradation includes contractile proteins is uncertain, and the impact of fasting on muscle strength has not been thoroughly studied. Grip strength is well-preserved during the first two weeks of fasting, thereafter declining[4,5], while strength in the legs remains unexplored. Endurance capacity declines noticeably after just 24–72 h of fasting[6,7], even though the expression of mitochondrial oxidative enzymes in skeletal muscle has been reported to be unaltered after three days of fasting[8,9].

[1]Norwegian School of Sport Sciences, Oslo, Norway. [2]Steno Diabetes Center Odense, Odense University Hospital, Odense, Denmark. [3]Department of Biochemistry, University of Cambridge, Cambridge, UK. [4]Laboratory of Mitochondrial Biology and Metabolism, National Heart, Lung and Blood Institute, National Institutes of Health, Maryland, USA. [5]Department of Clinical Medicine, Aarhus University, Aarhus, Denmark. [6]August Krogh Section for Molecular Physiology, Department of Nutrition, Exercise and Sports, University of Copenhagen, Copenhagen, Denmark. [7]Department of Nutrition, Division for Molecular Nutrition, University of Oslo, Oslo, Norway. [8]The Rowett Institute, Foresterhill Health Campus, University of Aberdeen, Aberdeen, UK. [9]Section of Specialized Endocrinology, Department of Endocrinology, Oslo University Hospital, Oslo, Norway. [10]Faculty of Medicine, University of Oslo, Oslo, Norway. [11]MRC Metabolic Diseases Unit, Institute of Metabolic Science, University of Cambridge, Cambridge, UK. [12]These authors contributed equally: Emelie T. F. Nilsen, Steffen Brufladt. ✉e-mail: jorgen.jensen@nih.no

Glycogen stores are limited, and liver glycogen is completely depleted after 24–36 h without food[10,11]. In contrast, muscle glycogen, which is the main substrate during exercise of moderate and high intensity, decreases only by 20–30% after three days of fasting[8,12,13]. Muscle glycogen is also a main substrate for anaerobic energy production[14], required for high-intensity "fight or flight" activities. Knowledge about muscle glycogen regulation during fasting is, therefore, essential to understanding humans' capacity for physical effort during periods of starvation. In addition, fasting elicits some well-established changes in plasma metabolites (decreased plasma glucose and increased plasma fatty acids (FA) and ketones), which have been studied for up to 40 days of fasting under resting conditions[1,15]. A recent study found that the plasma concentration of unsaturated FAs increased much more compared to saturated FAs after ten days of fasting[16]. The corresponding metabolic responses during exercise after prolonged fasting remain unknown.

The aim of the present study was to address several unexplored aspects of prolonged fasting and its effects on physical performance and skeletal muscle adaptations. We report preservation of maximal strength in leg muscle, despite a significant loss of lean mass. Further, there was a marked decline in peak oxygen consumption after six days of fasting. We identified a 13-fold increase in pyruvate dehydrogenase kinase 4 (PDK4) expression, increased pyruvate dehydrogenase (PDH) phosphorylation and compromised carbohydrate oxidation during aerobic exercise, despite unchanged expression of oxidative enzymes and preservation of 50% of muscle glycogen content. Therefore, humans maintain their capacity for physical abilities well during periods of severe food shortage. However, whereas muscle strength was preserved, there was a 10–15% decrease in high-intensity endurance capacity despite the expression of mitochondrial enzymes remaining unchanged in skeletal muscle.

## Results

### Characteristics of participants

Of the fifteen initial participants included two discontinued fasting and were thus excluded from the study. Thirteen participants, comprising six females and seven males, completed a seven-day water-only fasting intervention (Fig. 1A). Of note, the present data are from a different cohort than our previous publication[17]. Participants completed questionnaires and were clinically evaluated daily. No serious adverse events were reported during the fasting. The participants were young, healthy, without significant medical history, and did not use any prescription drugs or tobacco products. Baseline anthropometric data, age, and peak oxygen consumption ($\dot{V}O_{2peak}$) are presented in Table 1.

### Change in glucose, body mass and composition

Using Continuous Glucose Monitoring (CGM), we observed a decline in glucose levels from day zero to day three, subsequently stabilising (Fig. 1B). The average body weight decreased by $5.8 \pm 0.3$ kg (Fig. 1C), and in relative terms by $7.5 \pm 0.3\%$ after seven days of fasting (Fig. 1E). Correlation analyses revealed that the initial body weight was associated with reduction in body weight ($r = 0.62$, $p = 0.023$). Through Dual-Energy X-ray Absorptiometry (DXA) analyses, we quantified the alterations in lean, fat, and bone mass (Fig. 1D). After seven days of fasting, lean mass diminished by $4.6 \pm 0.3$ kg, an 8.0% decrease. We subsequently analysed the extremities as lean mass lost could influence physical tests. This revealed a $1.7 \pm 0.5$ kg reduction in lean mass across arms and legs, equivalent to a 6% reduction in both arms and legs (Fig. 1E). Fat mass in the extremities was reduced by $5.7 \pm 1.0\%$. Total fat mass was reduced by $1.4 \pm 0.1$ kg ($8.4 \pm 1.0\%$; Fig. 1D). The reduction in fat mass was associated with the body weight ($r = 0.61$, $p = 0.03$) and lean mass ($r = 0.73$, $p = 0.005$) prior to fasting. The Total bone mass remained unchanged throughout the fasting period (Fig. 1D). Detailed changes in body composition for both sexes in response to seven-day fasting are provided in Supplementary Table 1.

Protein loss was ascertained by analysing nitrogen excretion in urine. The daily nitrogen excretion exhibited a steady decline from approximately 15 g/day to approximately 10 g/day by the end of the fasting week (Fig. 1F), accumulating to $83.9 \pm 6.7$ g of nitrogen. Using a nitrogen factor of 6.25, the lost urine nitrogen corresponded to $524 \pm 42$ g of protein, and assuming a protein content of 20% of cellular

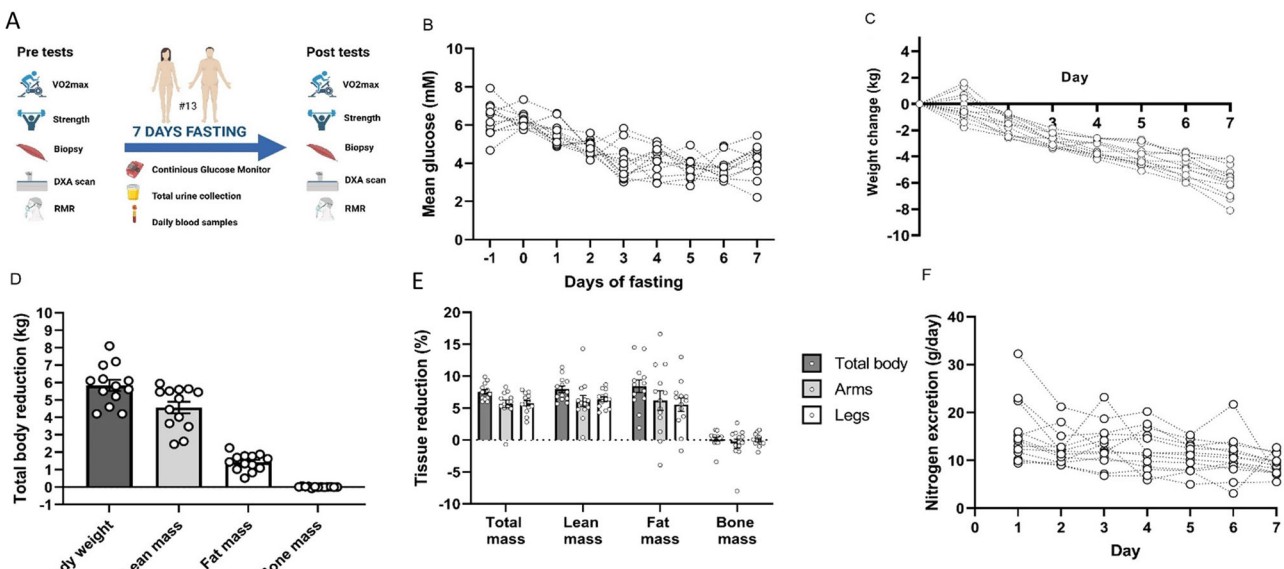

**Fig. 1 | Study designs, interstitial glucose, changes in body weight and body composition, and nitrogen excretion during fasting. A** Schematic representation of the study design illustrating phases of data collection. **B** Continuous measurement of glucose concentration (Dexcom). Data are reported as means from 8 a.m. to 8 a.m. the following day (n = 12). **C** Change in bodyweight throughout the week, in absolute terms in kilograms. **D** Reduction in total body, lean, fat and bone mass during the seven days' fasting calculated from DXA scan, in absolute values (kg). **E** Reduction total, lean, fat and bone mass of extremities and total body during the fasting period, relative to baseline values (%). **F** Daily nitrogen excretion during the fasting period calculated from total urine. Bar diagrams show means ± SEM (Fig. **D** and **E**). Figure 1A: "Created in BioRender. Møller, P. (2025) https://BioRender.com/z88l355. Data are from 13 participants except for glucose (n = 12).

## Table 1 | Baseline Characteristics

| | All ($N = 13$) | Males ($N = 7$) | Females ($N = 6$) | P-value |
|---|---|---|---|---|
| Age (years) | 29.7 ± 1.7 | 31.0 ± 1.6 | 28.1 ± 2.9 | 0.418 |
| Height (cm) | 177 ± 3 | 184 ± 3 | 169 ± 2 | <0.007 |
| Weight (kg) | 79.6 ± 5.0 | 92.3 ± 5.6 | 64.8 ± 2.6 | <0.002 |
| BMI | 25.0 ± 0.9 | 27.2 ± 0.9 | 22.6 ± 1.0 | <0.006 |
| Lean mass (kg) | 58.3 ± 4.1 | 68.90 ± 4.3 | 45.9 ± 2.3 | <0.001 |
| Body fat (kg) | 18.6 ± 1.9 | 20.6 ± 2.8 | 16.4 ± 2.3 | 0.278 |
| Body fat (%) | 23.4 ± 1.9 | 22.0 ± 2.3 | 25.1 ± 3.1 | 0.438 |
| Visceral fat (g) | 463 ± 148 | 763 ± 212 | 114 ± 77 | 0.021 |
| Bone mass (g) | 3159 ± 168 | 3597 ± 165 | 2649 ± 107 | <0.001 |
| VO$_{2peak}$ (ml/kg/min) | 47.9 ± 1.9 | 47.4 ± 2.0 | 48.4 ± 3.6 | 0.799 |
| VO$_{2peak}$ (L/min) | 3.77 ± 0.22 | 4.34 ± 0.18 | 3.11 ± 0.19 | <0.001 |

Data are means ± SEM, for all (13), males (7) and females (6). *P*-values are for differences between males and females (two-sided *t* tests). Abbreviations; BMI: Body mass index; VO$_{2peak}$: Peak oxygen uptake.

Baseline characteristics of the participants: Anthropometrics, body composition (lean, fat and bone mass) from DXA scan and peak oxygen uptake ($\dot{V}O_{2peak}$) was tested on an ergometer bike.

mass, this, in turn, corresponds to ~ 2.6 kg lean mass. Nitrogen excretion correlated with baseline lean mass ($r = 0.85$, $p < 0.001$).

### Physical performance

Maximal knee extensor strength remained unaltered after six days of fasting. This encompassed no changes in isometric strength (Fig. 2A), isokinetic peak torque across all three contraction velocities (Fig. 2B), or mean power output (Fig. 2C).

Results from the cardiopulmonary exercise test are detailed in Table 2. $\dot{V}O_{2peak}$ showed a 13% decrease in absolute terms, with a more modest reduction of 7% when normalised to body weight (Fig. 2D, E). Concurrently, the peak power output during the test diminished by 16%; Fig. 2F). Reduction in peak power output relative to body weight is shown in Fig. 2G. HR$_{peak}$ was slightly higher before the fasting period compared to after prolonged fasting (Table 2). The O$_2$ pulse decreased by 12% after fasting ($p < 0.00001$). Furthermore, peak blood lactate concentrations (~ 11 mM) and RPE achieved similar levels both before and after six days of fasting. Conversely, the Respiratory Exchange Ratio (RER) during exercise was 1.12 pre-fasting but only reached 0.93 post-fasting (Table 2).

### Metabolism at rest

Using indirect calorimetry, resting $\dot{V}O_2$ and resting metabolic rate (RMR) showed no changes on day five of fasting compared to before the prolonged fasting period (Fig. 3A, B). Resting RER declined from 0.86 ± 0.02 to 0.76 ± 0.01 ($p = 0.0004$; Fig. 3C) and fat oxidation increased its contribution from 37 ± 6% to 73 ± 3% ($p < 0.001$) of total energy turnover (Fig. 3D), while carbohydrate contribution fell from 53 ± 6% to 19 ± 3% ($p < 0.001$, $n = 13$). Blood pressure was lower after five days of fasting compared to before fasting (Systolic: 120 ± 5 vs 131 ± 6 mmHg, $p < 0.002$; Diastolic: 73 ± 2 vs 80 ± 3 mmHg, $p < 0.005$, $n = 13$) and resting capillary lactate was higher in the fasted state (1.5 ± 0.2 vs 1.1 ± 0.1 mM, $p < 0.05$, $n = 13$).

### Metabolism during fat oxidation test

During the maximal fat oxidation test, $\dot{V}O_2$ gradually increased during exercise with a similar pattern before and after fasting, but was slightly higher at three time points after the prolonged fasting period (Fig. 3E). RER was lower at every incremental step after the fasting period (Fig. 3F), in line with increased fat oxidation rates and decreased carbohydrate oxidation (Fig. 3G, H). The maximal fat oxidation rate increased from ~ 0.4 g/min after overnight fasting, to almost 0.8 g/min

after six days of fasting ($p < 0.001$; Fig. 3I), which occurred at similar absolute work rates before and after fasting. However, the intensity at which the maximal fat oxidation rate was reached relative to VO$_{2peak}$ ("Fat max"), increased from 46 ± 2% to 60 ± 3% of $\dot{V}O_{2peak}$ ($p < 0.001$; Fig. 3J). Although lactate concentration was higher at two time points during the test in the fasting state, the dynamics showed a similar pattern (Fig. 3K).

### Metabolic response to exercise

To investigate the metabolic response during exercise, blood samples were obtained before exercise, after the maximal fat oxidation test (highest load: 3 min at 80% of $\dot{V}O_{2peak}$) and after the $\dot{V}O_{2peak}$ test (at exhaustion) both before and after prolonged fasting. Before fasting, plasma glucose was similar at rest and after 80% $\dot{V}O_{2peak}$, but increased from 5.4 ± 0.2 to 6.7 ± 0.4 mM at $\dot{V}O_{2peak}$ ($p < 0.001$). After six days of fasting, plasma glucose at rest had decreased to 3.8 ± 0.2 mM ($p < 0.001$) compared to before fasting, but increased to 5.0 ± 0.2 mM ($p < 0.001$) after the maximal fat oxidation test, and further increased to 5.6 ± 0.2 mM ($p < 0.05$) at $\dot{V}O_{2peak}$ (Fig. 4A). Similar patterns were observed with plasma insulin (Fig. 4B). Plasma β-Hydroxybutyrate was low before the prolonged fasting period (0.07 ± 0.03 mM), both at rest and in response to exercise (Fig. 4C). After six days of fasting, the plasma concentration of β-hydroxybutyrate was 4.01 ± 0.30 mM before exercise (Fig. 4C). Interestingly, β-hydroxybutyrate decreased to 3.17 ± 0.27 mM after the maximal fat oxidation test, and further declined to 2.90 ± 0.24 mM after the $\dot{V}O_{2peak}$ test ($p < 0.001$).

### Metabolomics

In response to prolonged fasting, the plasma FA concentration nearly quadrupled (~ 0.4 vs ~ 1.5 mM; Fig. 4D). Metabolomics analysis was performed to investigate the individual constituents of FA (Supplementary Fig. 1). The most abundant FAs in human plasma, 16:0 (palmitic acid) and 18:1 (oleic acid), showed divergent responses. 16:0 only increased by ~ 21%, while 18:1 increased three-fold (Supplementary Fig. 1C and Fig. 1G). Generally, the percentage increase in saturated FAs was much lower compared to that of unsaturated ones. Notably, all measured fatty acids, regardless of their chain length and saturation, were significantly elevated in the fasting state at one time point during the test; either before exercise, after the maximal fat oxidation or the $\dot{V}O_{2peak}$ tests. Exercise did not significantly affect the total plasma FA concentration, or any of the individual FAs before or after the prolonged fasting (Fig. 4D and Supplementary Fig. 1).

Metabolomics data on amino acids (AAs) and the effect of fasting and exercise are presented in Supplementary Fig. 2. The AAs were categorised into branched-chained amino acids (BCAA), glucogenic and gluco- and ketogenic amino acids. Before fasting, exercise did not affect the plasma levels of any glycogenic AAs. At rest, all BCAAs were elevated in response to prolonged fasting as expected[15], but only leucine was significantly altered during the exercise test showing a decline of ~ 10%. Concurrently, isovaleric acid, a BCAA degradation product, was elevated after prolonged fasting but remained unchanged after the exercise tests. After prolonged fasting, L-alanine was the only glycogenic amino acid to increase in response to exercise, while L-serine was the only one to decrease. AAs possessing both ketogenic and glycogenic attributes were unchanged in response to the prolonged fasting, with the exception of tryptophan, which was reduced, but still unchanged during exercise.

### Muscle biopsies

Muscle glycogen content in *m. vastus lateralis* was 408 ± 34 mmol/kg protein before the fasting period, but declined to 191 ± 13 mmol/kg protein following the fasting period (Fig. 5A). Total glycogen synthase activity was unchanged in response to the week-long fasting (Fig. 5B). However, the fractional activity of glycogen synthase was higher after fasting compared to before the fasting period (Fig. 5C).

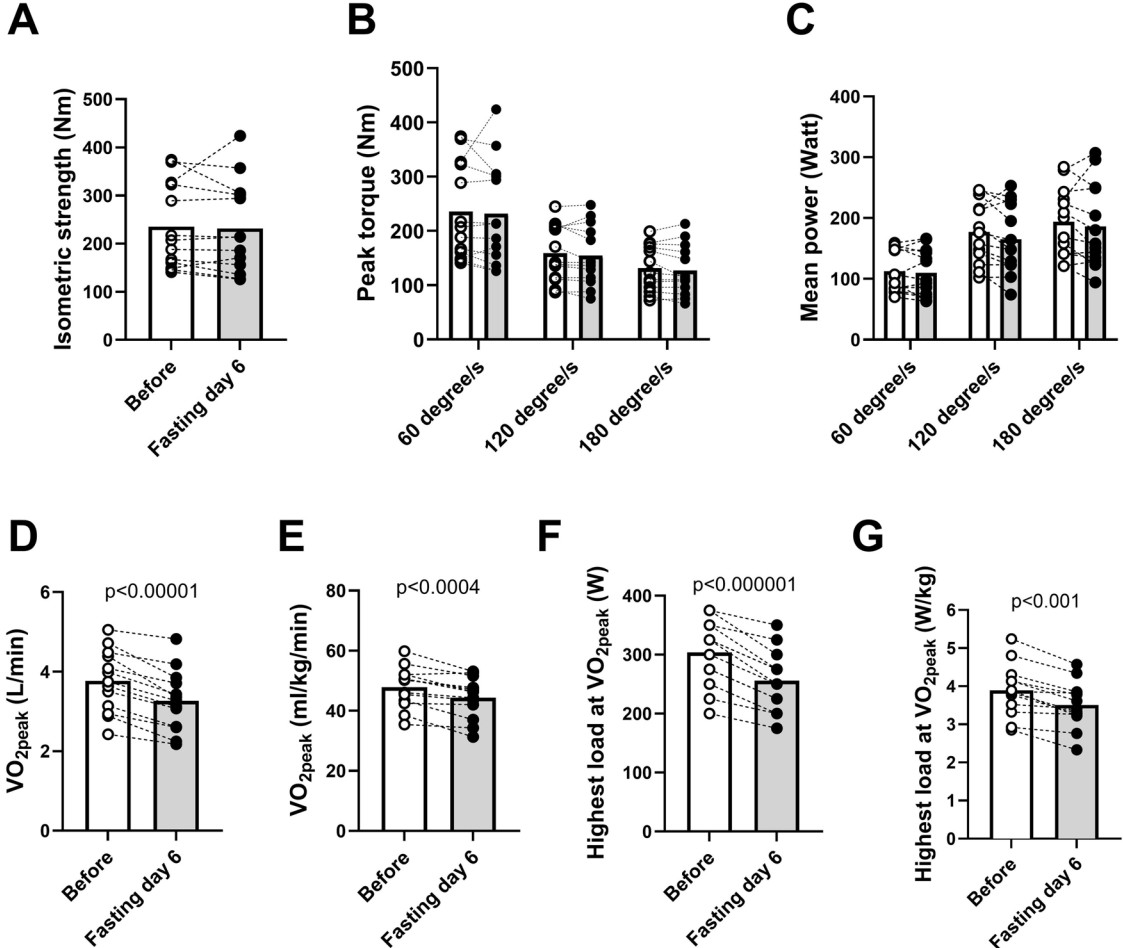

**Fig. 2 | Muscle strength and peak oxygen consumption tests before and after six days' fasting.** Results from isometric and isokinetic strength testing and $\dot{V}O_{2peak}$ tests before and after six days of fasting. **A** Isometric strength in newton metres (Nm). **B** isokinetic peak torque at three different velocities in Nm. **C** Mean isokinetic power at three different velocities (**D**) $\dot{V}O_{2peak}$ in absolute terms; L/min (**E**) $\dot{V}O_{2peak}$ related to body weight; ml/kg/min. **F** Peak power output (watt) during $\dot{V}O_{2peak}$ test. **G** Peak power output during the $VO_{2peak}$ test related to body weight. White and grey bars represent means before and after six days of fasting, respectively. Bars show means. White circles represent individuals before fasting and black after fasting, with dotted lines to illustrate individual change. Two-sided paired t-tests were used for comparisons. $N = 13$.

Phosphorylation of glycogen synthase at sites 1a and 2 + 2a remained unchanged (Fig. 5D, E), while phosphorylation at sites 3a + 3b was reduced post-fasting (Fig. 5F).

We conducted a detailed examination of selected muscle proteins known for their roles in mitochondrial function and the metabolism of glucose and lipids. The activity of AMPK was assessed across its three major complexes in skeletal muscles[18]; α1β2γ1, α2β2γ1, and α2β2γ3, and no discernible changes were observed post-fasting (Fig. 6A–C). The protein content of pyruvate dehydrogenase (PDH) remained unaffected by prolonged fasting (Fig. 6D). The inhibitory phosphorylation of PDH was enhanced at both sites 1 and 2, which suppresses PDH activity (Fig. 6E, F). PDK4, the kinase responsible for the inhibitory phosphorylation of PDH, displayed minimal expression in muscle tissue before the prolonged fasting. After prolonged fasting, we observed a 13-fold increase in PDK4 expression (Fig. 6G and Supplementary Fig. 3).

Skeletal muscle expressions of hexokinase II, citrate synthase, and components of the mitochondrial respiratory complexes I-V were unaffected after the prolonged fasting (Supplementary Table 2 and Supplementary Fig. 4). Fatty acid transport protein 4 (FATP4) and acetyl-CoA carboxylase (ACC), both involved in lipid metabolism, also remained unchanged in response to the prolonged fasting (Supplementary Table 2).

## Discussion

Our study's main finding is that maximal leg muscle strength was preserved during fasting, despite a significant loss of lean body mass. In addition, we found a marked decline in peak oxygen consumption after six days of fasting. We observed an increase in alanine and a decrease in leucine during exercise after prolonged fasting, whereas the exercise protocol caused no changes before fasting. Similar changes in glucose and amino acids are known to occur during more prolonged exercise in the fed state[19], suggesting that the processes responsible for altered substrate availability during exercise are more rapidly activated after prolonged fasting. In addition, both the clinical and paraclinical findings were consistent and did not differ between sexes, in agreement with our previous study[17]. After fasting, PDK4 protein content increased 13-fold and inhibitory phosphorylation of PDH was elevated. This regulatory step diminishes skeletal muscles' ability to oxidise carbohydrates and might be necessary to prevent hypoglycaemia. However, it represents a compromise since it also reduces the maximum capacity during aerobic exercise. It is possible that this represents the primary mechanism underlying the diminished aerobic exercise capacity after prolonged fasting.

Participants exhibited a 7.5% reduction in body weight over the week, aligning with previous studies[2,4,20]. In relative terms, the loss of fat and lean mass were similar and uniformly distributed across the

**Table 2 | Physiological and metabolic responses to the $\dot{V}O_{2peak}$ test**

| | Before | 6 Days fasting | *P*-value |
|---|---|---|---|
| VO$_{2peak}$ (L/min) | 3.77 ± 0.22 | 3.27 ± 0.21 | <0.0001 |
| VO$_{2peak}$ (ml/min/kg) | 47.9 ± 1.9 | 44.4 ± 1.9 | 0.0004 |
| Load (Watt) | 304 ± 15 | 256 ± 14 | <0.00001 |
| Lactate (mM) | 11.1 ± 0.6 | 10.5 ± 0.8 | 0.506 |
| RPE (Borg scale) | 19.2 ± 0.3 | 19.5 ± 0.2 | 0.219 |
| Peak heart rate (bpm) | 186 ± 3 | 181 ± 2 | <0.01 |
| RER (VCO$_2$/VO$_2$) | 1.12 ± 0.01 | 0.93 ± 0.02 | <0.000001 |
| Ventilation (L/min) | 154 ± 10 | 140 ± 12 | 0.011 |
| Ventilation/VO$_2$ (L/L) | 41.0 ± 1.3 | 42.4 ± 1.9 | 0.180 |
| O$_2$ Pulse (mL/beat) | 20.4 ± 1.3 | 18.1 ± 1.2 | <0.0001 |

Data are means ± SEM. Two-sided paired t-tests were used for comparisons. N = 13. Abbreviations; VO$_{2peak}$: Peak oxygen uptake; RPE: Rate of perceived exertion; RER: Respiration exchange ratio.

Peak oxygen uptake ($\dot{V}O_{2peak}$) and connected parameters during the test. $\dot{V}O_{2peak}$ was tested with an incremental protocol (increase 25 W/min) where participants cycled until exhaustion. Oxygen uptake was measured continuously, and $\dot{V}O_{2peak}$ was calculated as the mean of the two highest consecutive measurements over 30 s. Capillary lactate was measured 1 min after exhaustion. The rate of perceived exertion (RPE) was scored according to the Borg scale. Heart rate was measured continuously, and the highest (peak) value was reported (HF$_{peak}$). The data on respiratory exchange ratio (RER) and ventilation are means from the period of measurements of $\dot{V}O_{2peak}$.

body, without any effect on bone mass. The loss of body fat, at 1.4 kg, corresponds to a daily energy supply of 7530 kJ (1800 kcal), nearly sufficient to supply the measured RMR. There was no decrease in RMR after five days of fasting, but substrate oxidation changed towards fat oxidation as expected[20] and in agreement with the available substrates.

We investigated the effect of prolonged fasting on strength in leg muscles, which was critical for movement and survival in early humans. Grip strength has mostly been investigated previously and seems to be preserved during the first two weeks of fasting[4,5,21]. To our knowledge, Knapik et al. are the only investigators who have studied dynamic strength during fasting, and they reported a 10% reduction in isokinetic elbow flexion after three and a half days of fasting, without decreased handgrip strength[21]. Our investigation of isokinetic strength at three velocities and isometric strength showed that knee extensor strength was preserved in all tests after prolonged fasting, despite a 6% reduction in lean mass of the legs. Skeletal muscle protein is commonly regarded as the primary source of amino acids during periods of stress such as illness and starvation[3,22], but it is unclear whether the degraded proteins originate from intracellular contractile proteins or other sources. The finding that muscle strength is well preserved after six days of fasting in young, healthy adults argues against severe breakdown of contractile elements. The finding of preserved strength during fasting is in stark contrast to a week of bed rest, where a 5–10% decline in strength has been reported, even without a large loss of lean mass[23]. The nature of the degradation of specific proteins during fasting was not resolved, and the degradation of myosin and actin most likely determines the decline in force. Importantly, collagen comprises 25–30% of the total protein in the human body[24], and degradation of non-contractile proteins may occur without a decline in muscle strength. Our observation underlines the robustness of the human body and its ability to maintain strength in major muscles of locomotion under nutrient-deprived conditions when allowed activity.

The effect of prolonged fasting on aerobic capacity has been extensively studied from both military and performance perspectives, using various exercise protocols, since the 1950s[25]. Aerobic capacity has been reported to drop after only 24 h of fasting[6,7]. In the present study, peak watt decreased by 16% and $\dot{V}O_{2peak}$ was reduced by 13% after fasting, which is comparable to studies on fasting for 5–10 days[5,25].

Here, we report a dramatic 13-fold increase in the protein content of PDK4 in human skeletal muscle after prolonged fasting. PDK4 is responsible for the phosphorylation and inhibition of PDH, regulating the entry of pyruvate into the citric acid cycle[26], and thereby carbohydrate oxidation in skeletal muscles[27]. This maintains blood glycemia during fasting, exemplified by mice lacking PDK4 becoming severely hypoglycemic when fasting, whereas PDK4 expression increases in normal mice when fasting[28]. PDH activity is reduced by more than 50% in human muscle after 72 h fasting[9], in accord with our findings of increased inhibitory phosphorylation of PDH after fasting. This inhibition of carbohydrate oxidation might be a prerequisite to avoid hypoglycemia during fasting, but may also prevent humans from increasing aerobic energy turnover during exercise. Our results show no changes in the expression of mitochondrial respiratory complexes or enzymes such as HK2 and citrate synthase. The finding of preserved mitochondrial enzymes combined with the increased inhibitory phosphorylation of PDH, suggests that the downregulation of carbohydrate oxidation is at least in part due to increased PDK4 expression.

Muscle glycogen is an important energy substrate for both aerobic and anaerobic exercise. The glycogen content in skeletal muscle was reduced by ~50% during fasting. This is in agreement with earlier studies showing a 20–30% reduction of glycogen after two to three days of fasting[8,13,29]. This level of glycogen, while lower, is still sufficient to sustain anaerobic capacity[14], supporting the body's ability to maintain "fight or flight" responses during starvation. The preservation of anaerobic capacity is additionally supported by the observation of similar lactate dynamics before and after fasting, indicating intact glycolytic capacity. The fact that RER only reached 0.93 during the exhaustive $\dot{V}O_{2peak}$ test after prolonged fasting, despite a reasonable amount of available muscle glycogen and high plasma lactate concentration, highlights how efficient the inhibition of carbohydrate oxidation is during exercise in the fasting state.

AMPK is known for its pleiotropic effects on cell metabolism[30], and studies in mice suggest that it stimulates protein degradation in muscle[31]. AMPK phosphorylation in human skeletal muscle is unchanged during two to three days of fasting[12,32]. We investigated AMPK activity comprehensively in the three common muscle complexes and observed no effect of fasting on any of the complexes. Together with the observation of a gradual decrease in nitrogen excretion, this questions the role of AMPK as a major regulator of protein degradation in human skeletal muscle during fasting.

The exercise-induced metabolic responses in plasma glucose and AAs were more pronounced at lower intensities after fasting. Plasma glucose has been reported to be unaltered in response to exercise at low intensity after 21 days of fasting[33], but increased after 39 days of fasting[34]. In the present study, plasma glucose levels increased during the maximal fat oxidation test after fasting and further increased during the $\dot{V}O_{2peak}$ test. Compared to before fasting, we only observed an increase in plasma glucose at $\dot{V}O_{2peak}$ as expected[35].

We also investigated the effect of exercise on plasma AAs after fasting, and the exercise protocol resulted in a decline in the BCAA leucine and an increase in alanine after fasting. Exhaustive exercise decreases plasma BCAA and increases alanine in the fed state[19,36]. The changes in plasma glucose and AAs in response to exercise are generally consistent between fasting and fed states, but the magnitude of these changes is greater, and they occur at lower exercise intensities after fasting.

Plasma FA increased to ~1.5 mM after three days of fasting and plateaued, until six days of fasting. The maximal rate of fat oxidation nearly doubled after fasting. Despite these changes, there was no increase in the expression of FATP4 or ACC in skeletal muscle, similar to a previous study after fasting for three days[9]. During prolonged

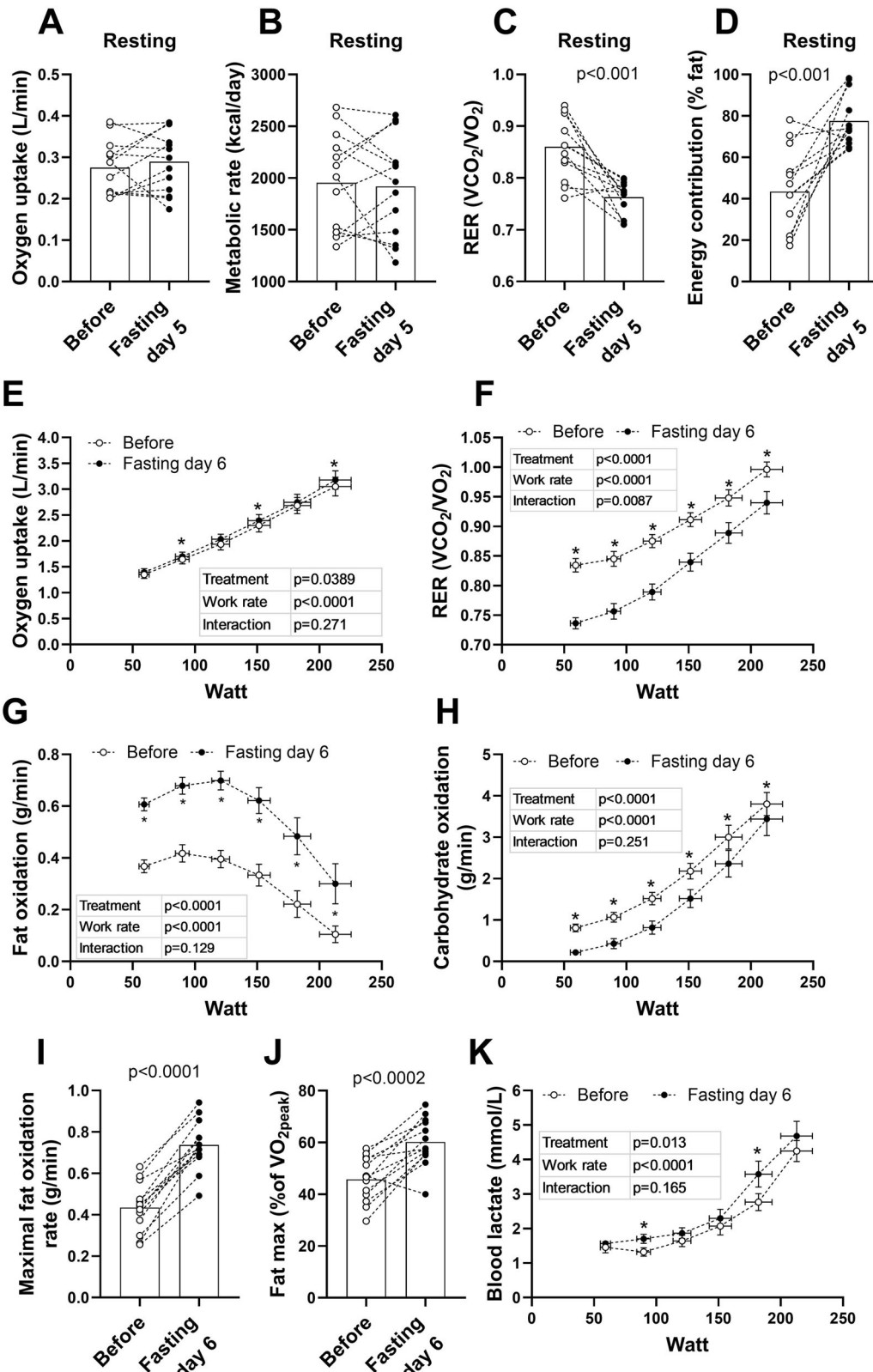

**Fig. 3 | Metabolism at rest and during maximal fat oxidation test.** Metabolism, energy expenditure and substrate selection at rest and during the incremental fat oxidation test: **A** Resting oxygen consumption. **B** Resting metabolic rate. **C** RER values during rest. **D** Energy contribution (% fat) during rest. **E** $\dot{V}O_2$ during incremental fat oxidation test. **F** RER during incremental fat oxidation test. **G** Fat oxidation during incremental fat oxidation test. **H** Estimated carbohydrate oxidation during incremental fat oxidation test. **I** Maximal fat oxidation during incremental fat oxidation test, (**J**) % $\dot{V}O_{2peak}$ where fat max was achieved during incremental fat oxidation test. **K** Blood lactate during incremental fat oxidation test. White and grey bars represent means before and after six days of fasting, respectively. Bars show means. Error bars show SEM. White circles represent individuals before fasting and black after fasting, with dotted lines to illustrate individual change. Two-sided paired *t* tests were used for comparisons in figures (**A–D** and **I–K**). Repeated ANOVA was used for statistical analyses in figures (**E–H** and **K**), with LSD as a post hoc test. *: difference between pre and post. *N* = 13. Abbreviations: RER: respiratory exchange ratio, RMR: Resting metabolic rate.

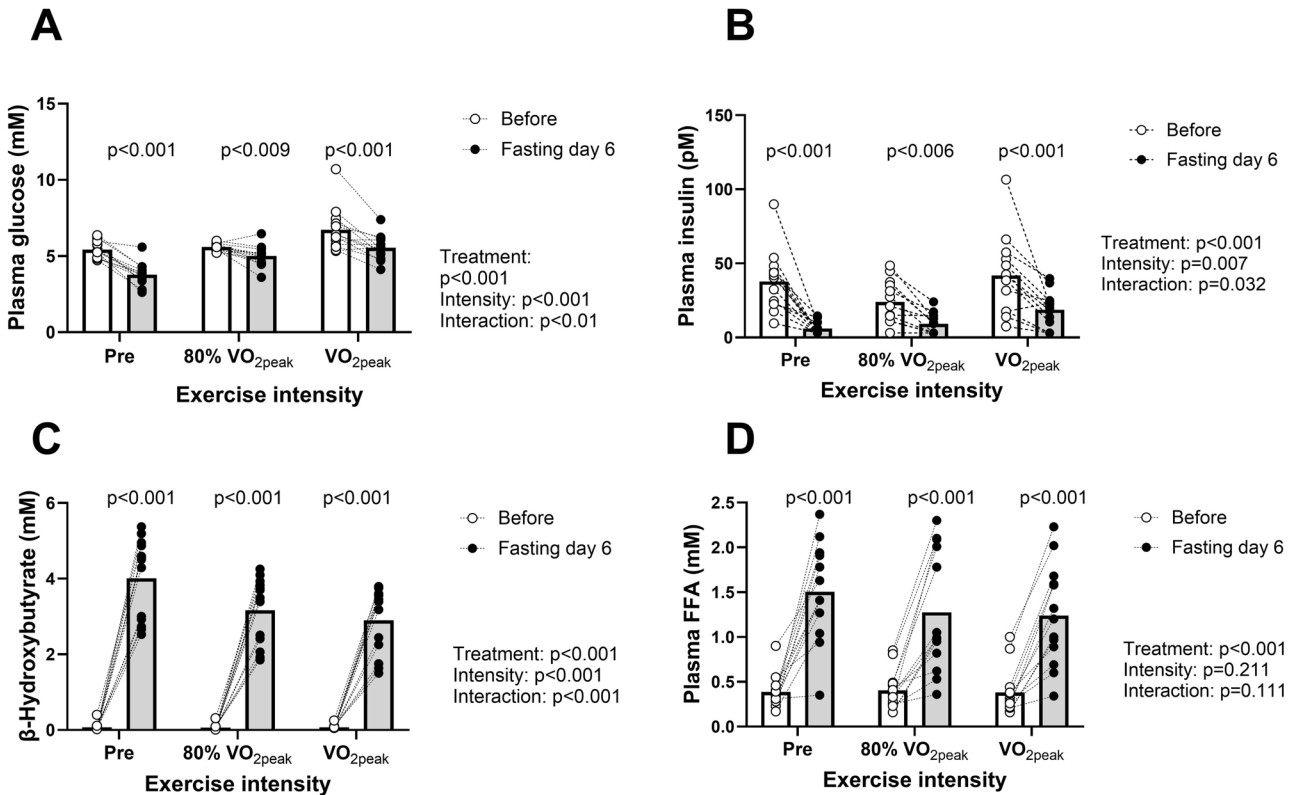

**Fig. 4 | Plasma metabolites at rest and during exercise.** Plasma metabolites before exercise, after maximal fat oxidation test and $\dot{V}O_{2peak}$ test, before (white bars) and after six days' fasting (grey bars). White circles represent individuals before fasting and black after fasting, with dotted lines to illustrate individual change: Plasma levels of (**A**) glucose, (**B**) insulin, (**C**) β-hydroxybutyrate, and (**D**) free fatty acids (FAs). Repeated ANOVA with LSD as post hoc tests were used for statistical analyses. $N = 13$.

exercise in the fasting state, plasma FA becomes the dominant energy substrate in skeletal muscle[37]. The exercise test in our study lasted ~ 25 min and did not significantly change plasma FA before nor after fasting. Interestingly, Steinhauser et al. recently reported that plasma levels of unsaturated FAs increased twice as much as saturated FAs after ten days of fasting[16], as we also observed. This is in contrast to prolonged physical activity, where unsaturated and saturated FAs increase similarly[38]. We did not observe any significant changes in the concentrations of any of the FAs with our exercise protocol.

Ketones changed markedly in response to exercise in the fasting state. Ketone bodies serve as a key energy substrate during fasting, especially for the brain[1,39,40], but both skeletal and heart muscles oxidise ketones under hyperketonemia[37,41]. From the present cohort, we have previously reported a gradual increase in resting β-hydroxybutyrate (BHB)[42], where plasma BHB was almost absent after fasting overnight, but increased to 4 mM after prolonged fasting. In the present study, there was a 28% decrease in BHB during exercise in the fasting state. This finding is in line with earlier reports[37,43] supporting the proposition that BHB serves as an energy substrate in muscle. Importantly, ketones inhibit lipolysis in adipose tissue[44], and a decline in plasma ketones might be necessary to achieve a high rate of lipolysis in adipose tissue during exercise after prolonged fasting.

Strength and limitations: Dynamic muscle strength in leg muscles after prolonged fasting has never been investigated before, and we expanded the duration for investigation of glycogen, expression of metabolic enzymes, and AMPK activation in skeletal muscle from three to seven days of fasting. It is a strength that we included healthy participants with training experience and used state-of-the-art methodologies for physiological tests after familiarisation tests. Furthermore, we tested both strength and endurance in the same

cohort to ensure that the study could evaluate a holistic perspective of physical capacity. It is a strength that changes, both in clinical and metabolic variables, were consistent in all participants, making the data reliable. We consider it a strength that the study was performed in free-living conditions rather than on hospitalised participants, to imitate real-world scenarios. Moreover, we have demonstrated the feasibility of conducting weeklong fasting studies under free-living conditions with CGM, collection of urine, daily blood sampling, weighing and questionnaires. The questionnaires confirmed that the volunteers (unpaid) were motivated to continue, and the increase in β-hydroxybutyrate and FFA, in combination with the decline in weight and blood glucose, show that the participants followed the fasting regime. There were only two drop-outs early in the fasting period and no serious side effects. It is also a strength that we investigated both sexes.

A limitation of the present study is the lack of a randomised control group, which limits causal inference. Moreover, muscle biopsies were not collected before and after exercise, preventing us from isolating the effect of prolonged fasting and exercise on glycogen metabolism and enzymatic activation of relevant proteins. Although 13 participants completed the study, the sample size was relatively small. All participants were healthy, young adults, limiting the generalisability of the results. Furthermore, all tests were done at the same time of day, restricting us from investigating the potential effect of circadian rhythms during fasting. Despite participants doing the study in free-living conditions, today's free-living conditions are far from free-living conditions in the past. Future research with more diversified cohorts is necessary to substantiate these findings, particularly concerning the potential therapeutic effects on various diseases. Finally, the endurance testing protocol could have been more wide-ranging to

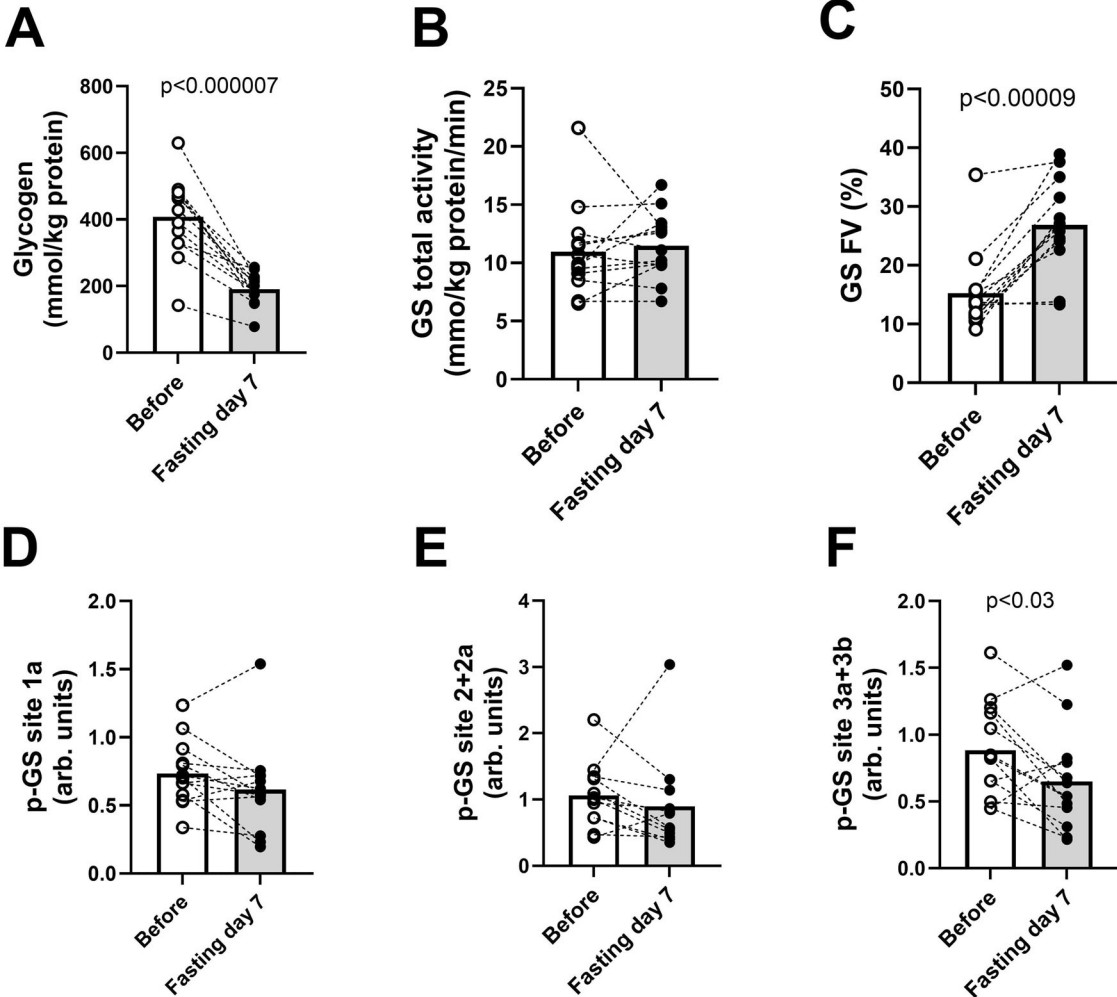

**Fig. 5 | Glycogen content and activity and phosphorylation of glycogen synthase.** Effect of seven days' fasting on glycogen content and glycogen synthase activation. White and grey bars represent means before and after seven days of fasting. White circles represent individuals before fasting and black after fasting, with dotted lines to illustrate individual change. **A** Intramuscular glycogen content (mmol/kg protein). **B** Total activity of glycogen synthase (mmol/kg protein/min). **C** Glycogen synthase fraction velocity (%), phosphorylation of glycogen synthase at site (**D**) 1a, (**E**) 2 + 2a, **F**: 3 + 3b. Bars show means. Abbreviation: GS: glycogen synthase, p: phosphorylation, FV: fraction velocity. Two-sided paired t-tests were used for comparisons. $N = 13$ except (**E**) where $N = 12$.

characterise the metabolic response during exercise more extensively. The inclusion of blood volume measurements would allow for estimating the effect of this on aerobic capacity.

In conclusion, maximal dynamic and isometric strength in the leg extensors did not deteriorate after six days of fasting, despite participants losing 4.6 kg of lean mass. In contrast, maximal oxygen uptake and work capacity decreased by ~15% without any significant decline in the expression of oxidative enzymes in skeletal muscle and only a 50% reduction in glycogen. We suggest that the 13-fold increase in PDK4 expression and increased inhibitory phosphorylation of PDH in skeletal muscle is the mechanism that prevents the muscle from oxidising carbohydrates during exercise in the fasting state. Furthermore, we suggest that reduced capacity for carbohydrate oxidation is responsible for the reduction in maximal oxygen uptake in the fasted state. The well-known changes in plasma metabolites during exercise after an overnight fast appear to be similar after prolonged fasting, but occur with greater amplitude and at lower intensities and durations of exercise. Skeletal muscle AMPK activity was not elevated in response to prolonged fasting, arguing against AMPK as a major driver for protein degradation in human skeletal muscle. From an evolutionary perspective, survival in periods without food has depended on the ability to search and hunt for food, and we showed that muscle

strength was preserved, but the ability to sustain intense aerobic work was markedly reduced.

## Methods

The study protocol was conducted in accordance with the Declaration of Helsinki. The study protocol was initially reviewed by the Regional Ethics Committee of Norway (2017/1052; REK sør-øst B) with the decision that the research project was outside the Act on Medical Health Research, confirmed in a letter of exemption (2017/1052b). The study received approval from the Ethics Committee at the Norwegian School of Sport Sciences (15-220817) and reported to the Norwegian Centre for Research Data (NSD: #327898).

### Study design
Prior to the fast, participants had no restrictions on their food intake. Baseline values for DXA scan, resting metabolic rate (indirect calorimetry), physical testing, and biopsies were taken after an overnight fast. Throughout the seven-day fasting period, participants were allowed only water consumption, but remained in normal free-living conditions, unrestricted from engaging in exercise and work. The resting metabolic rate was repeated on day five of fasting, and physical tests were repeated on day six. Biopsies and DXA scan were repeated in

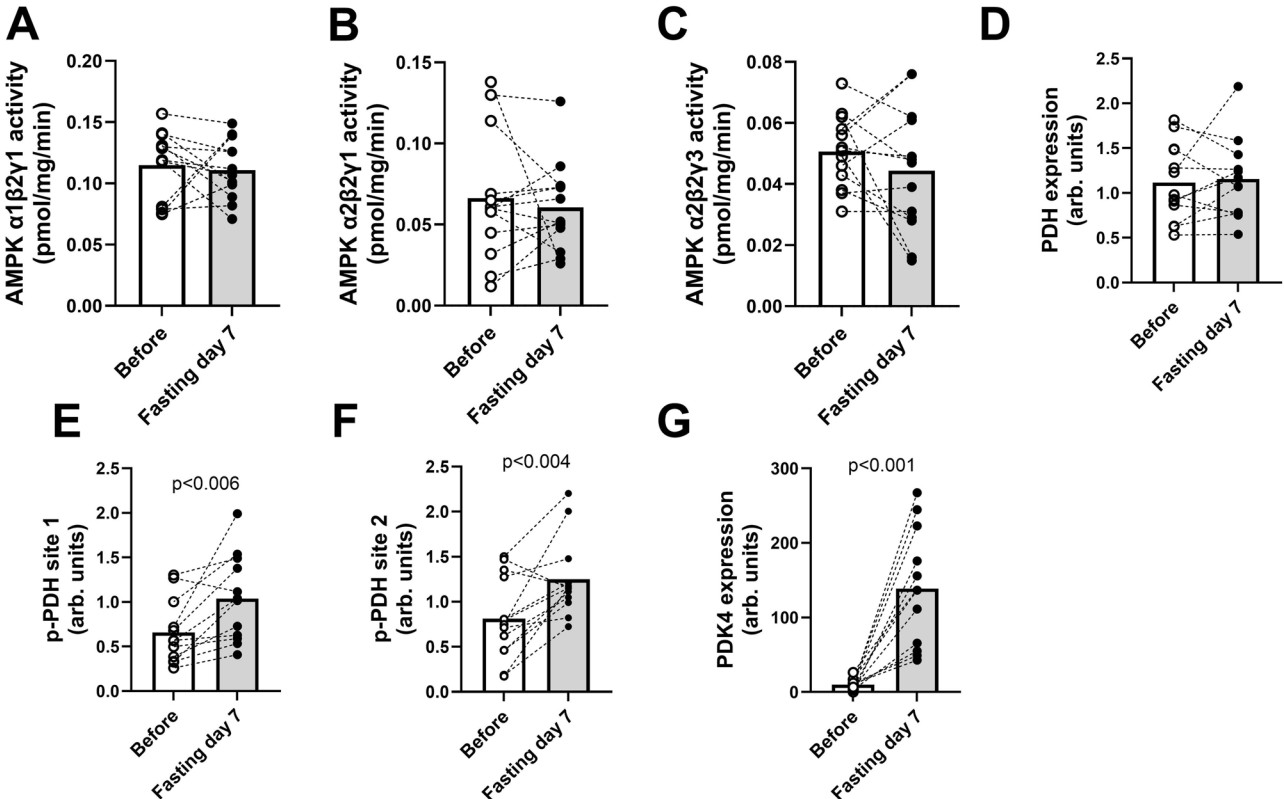

**Fig. 6 | AMPK activity, protein expression and phosphorylation of PDH and PDK4 before and after fasting.** Effect of seven days' fasting on activity, phosphorylation and expression of selected proteins. White and grey bars represent means before and after seven days of fasting. White circles represent individuals before fasting and black after fasting, with dotted lines to illustrate individual change. **A, B** and **C** Activity of AMPK complex: α1β2γ1, α2β2γ1, and α2β2γ3. **D** Protein expression of PDH. **E** Phosphorylation of PDH at site 1. **F** Phosphorylation of PDH at site 2. **G** Expression of PDK4. Bars show means. Two-sided paired *t* tests were used for comparisons. $N = 13$ for (**A**–**C**, **F** and **G**) $N = 12$ for (**D** and **E**). Abbreviation: AMPK: 5′-Adenosine monophosphate-activated protein kinase. PDH: pyruvate dehydrogenase, PDK4: pyruvate dehydrogenase kinase-4.

the morning on day seven of fasting. Participants met at the laboratory every morning (between 7 and 9 a.m.) for daily weighing, blood sampling, completion of questionnaires (See Supplementary Information) and delivery of urine. Glucose was measured continuously using a Dexcom G4 Platinum (Dexcom Inc., San Diego, CA, USA), and calibrated every morning and evening with capillary blood glucose (HemoCue Glucose 201 RT; Ängelholm, Sweden). For safety, we provided daily professional care during the intervention, side effect monitoring, clinical evaluation and equipped participants with continuous heart rate monitors. Participant recruitment was conducted through various methods, including the Norwegian School of Sport Sciences website, posters, and social networks. The sample size was calculated from being able to detect a decline in nitrogen excretion during the week of fasting. Participants who volunteered were screened before participation. Written and oral information regarding the project was provided, and signed consent was obtained before inclusion. Inclusion criteria for participants were as follows: healthy, aged 18–45 years, with a body fat percentage greater than 15% for females and 12% for males, and having some experience with exercise. Exclusion criteria: pregnancy or breastfeeding, use of tobacco products, any known disease (e.g., cardiovascular disease, diabetes or eating disorders) or use of any medication. The participants were unpaid and did not receive any compensation for participation.

### Anthropometrics and DXA scan
Body weight was measured with a precision of 0.1 kg using the Seca 877 scale (Seca Gmbh, Hamburg, Germany). Height was measured with a portable stadiometer Seca 213 (Numed Holdings Ltd, Sheffield, UK). Body composition was determined by DXA after overnight fasting (Lunar iDXA; GE Healthcare, Madison, USA) and analysed using enCORE software (v18; GE Healthcare, Madison, USA). DXA scans were performed twice before the fasting period (day -4 and -1, with Pre-data given as mean) and after seven days of fasting. Total body composition and regional analyses for legs and arms were assessed by adjusting the cut positions estimated by the enCORE software, with all scans being viewed side by side to ensure consistency. The cutline separating the arms from the trunk went through the glenohumeral joint. For the legs, one cutline was positioned medially between the legs and a second through the femoral neck and laterally to meet the iliac crest.

### Familiarisation with physical tests
Prior to the baseline tests, participants were familiarised with the dynamometer for knee extensor maximal isokinetic and isometric strength testing (Humac NORM Dynamometer; Computer Sports Medicine Incorporated, Stoughton, MA, USA). The equipment was appropriately adjusted, and the positions were documented to ensure consistency in all subsequent tests. After adjustment, participants completed the test as described below.

For endurance testing, participants underwent a familiarisation process with the ergometer bike (Lode Excalibur Sport, Groningen, Netherlands) prior to the intervention. The positions on the bike were documented to ensure consistency in all subsequent tests. During the familiarisation test, participants cycled for five minutes at 4 and 5 workloads with increasing intensities to establish the relationship between work rate and oxygen uptake ($\dot{V}O_2$). The initial work rate was set at 75, 100, or 125 W, depending on the participant's weight and training experience. At each load, $\dot{V}O_2$ was measured during the last 90 s (described below), and capillary lactate levels were assessed after

4.5 min (see below). The test was completed when blood lactate reached 4 mM. After that, the participants had three minutes' recovery before a 10 s maximal effort cycle test was performed, and after an additional three minute rest, $\dot{V}O_{2peak}$ was measured as described below. Linear regression was utilised to establish the relationship between load and oxygen uptake in order to calculate the load at the fat oxidation test.

## Physical tests

Physical tests were conducted in the morning (starting between 8 and 10 a.m. after resting blood samples were secured) after an overnight fast the day before the fasting period started and after six days of fasting. Participants first performed the strength tests, before the fat oxidation test, 10 s maximal effort test, and lastly, after an additional three minutes' rest, maximal oxygen uptake ($\dot{V}O_{2peak}$) was measured. Throughout all tests, the participants were provided with encouragement from the test leaders.

## Strength testing

Isokinetic and isometric strength were tested with a Humac dynamometer (Humac NORM, Computer Sports Medicine Incorporated (CSMi), Stoughton, MA, USA). The participants were seated in an upright position, and the chair was adjusted to have the rotational axis of the knee in the dynamometer's axis of rotation. The lever arm was secured just proximal to the ankle joint with a trap. The backrest (hip angle) was set at an angle of 85° and participants were secured with a four-point trap according to recommendations and kept their hands in their laps. Chair adjustments were noted and used in familiarisation, pre- and post-tests.

For a warm-up, participants cycled for five minutes, followed by three sets of sub-maximal effort at each angle velocity. Following the three warmup contractions, a 30 s rest period was provided before performing three maximal isokinetic knee extensions over 30 seconds. This procedure was repeated for all angle velocities: 60°•s⁻¹, 120°•s⁻¹ and 180°•s⁻¹ with 1 min rest between velocities. The maximal isometric strength was tested with a knee angle of 60°, preceded by a warm-up attempt, with three contractions of three seconds, with a one-minute rest between contractions. The tests with the highest peak torque from isometric strength, and peak torque and mean power from isokinetic strength are reported.

## Heart rate and indirect calorimetry

Fat oxidation and $\dot{V}O_{2peak}$ tests were performed with continuous heart rate measurements (Polar RS800CX, Polar Electro Oy, Kempele, Finland). The rate of perceived exertion (RPE) was rated by the Borg scale[45]. $\dot{V}O_2$ and carbon dioxide production ($\dot{V}CO_2$) were measured continuously by indirect calorimetry with a mixing chamber (Oxycon Pro; Jager Instr., Hoechberg, Germany), which was calibrated prior to each test. Volume calibration of the turbine (Triple V, Erich Jaeger GmbH, Hoechberg, Tyskland) was conducted with a 3 L pump (Calibration Syringe, series 5530, Hans Rudolph Instr; MO, USA) and the gas analysers were calibrated with a high-precision gas mixture of 6.0% $CO_2$ and 15.0% $O_2$ in nitrogen (Linde, Oslo, Norway).

## Fat oxidation

Maximal fat oxidation was assessed using a step-wise test with continuous measurement of $\dot{V}O_2$ and $CO_2$ production ($\dot{V}CO_2$), to evaluate the metabolic response to endurance exercise. After a 5-minute warm-up at 25% of $\dot{V}O_{2peak}$, the participants cycled for three minutes at intensities of 30, 40, 50, 60, 70 and 80% of $\dot{V}O_{2peak}$, maintaining a cadence of ~80 RPM. The mean values of $\dot{V}O_2$ and $\dot{V}CO_2$ during the final 60 s on each work rate were used to calculate fat oxidation, according to the formula: fat oxidation (g · min⁻¹) = 1.67 · $\dot{V}O_2$ (L · min⁻¹) − 1.67 · $\dot{V}CO_2$ (L · min⁻¹)[46]. Fat oxidation and power were fitted for each participant (third-order polynomial) for the determination of the

maximal rate of fat oxidation. For more detailed information about the calculation of the maximal fat oxidation rate and the intensity for maximal fat oxidation, see previous report[47].

## Peak oxygen uptake

The $\dot{V}O_{2peak}$ test was conducted using an incremental protocol, where the initial work rate was adjusted based on the initial test. The work rate increased by 25 W every 60 s until the participants reached voluntary exhaustion, or was terminated when they were no longer able to maintain a cadence of 60 RPM. $\dot{V}O_{2peak}$ was calculated as the mean of the two highest consecutive 30 s measurements. The highest heart rate recorded during the $\dot{V}O_{2peak}$ test was regarded as the peak heart rate (HF$_{peak}$).

## Resting metabolic rate

Participants arrived at the laboratory in the morning in a fasted state, having avoided exercise as a means of transportation. Resting metabolic rate (RMR) was measured two to three days before the prolonged fasting period started after an overnight fast and after five days of fasting by indirect calorimetry (Oxycon Pro) using the breath-by-breath technique in a quiet room with the participants lying comfortably on a mattress. RMR was measured between 6.00 and 8.00 a.m. before blood sampling and questionnaires. Gas exchange was measured over 20 min, with the equipment described above, using a face mask (7450 V2; Hans Rudolph Inc., Shawnee, KA, USA). The means of the respiratory data for the last five minutes were used to calculate RMR and substrate oxidation according to[46], and subsequently to calculate the non-protein resting metabolic rate assuming energy-to-substrate factors of 4 kcal/g (CHO) and 9 kcal/g (fat). Blood pressure was measured (WelchAllyn Spot Vital Signs LXi 0297, Skeneateles Falls, NY, USA) after RMR. Systolic and diastolic blood pressure were the means of two measurements. Finally, capillary lactate was measured.

## Blood and urine analyses

During exercise testing, a fingertip was washed and the skin punctured (Saft-T-Pro Plus, Accu-Check, Mannheim, Germany) before 50 µl capillary blood was collected. For measurement of lactate, 20 µl blood was injected into an YSI 1500 SPORT analyser (Yellow Springs Instruments Life Sciences, Yellow Springs, OH, USA). The analyser was calibrated with 5.0 mM lactate before each $\dot{V}O_{2peak}$ test.

Blood samples were taken each morning during the fasting period and during the exercise testing for the preparation of serum and plasma, using EDTA and Li-heparin tubes. Blood was taken in serum tubes (VACUETTE® serum tubes with separation gel; Greiner bio-one, Kremsmüster, Austria), coagulated for 30–45 min at room temperature, and then centrifuged (3500 × $g$ at 4 °C for 10 min). EDTA tubes (VACUETTE® EDTA-K2; Greiner bio-one) and lithium-heparin tubes (VACUETTE® Lithium-heparin tubes with separation gel; Greiner bio-one) were immediately placed on wet ice and centrifuged within 10 min (3500 × $g$ at 4 °C for 10 min). Plasma and serum were immediately pipetted in low bind Eppendorf tubes (Eppendorf 1.5 ml Lobind microcentrifuge tubes, Eppendorf, Hamburg Germany) and frozen on dry ice. All samples were stored at − 70 °C. Plasma insulin was measured with an enzyme-linked immunosorbent assay human insulin kit, K6219 (Dako, Glostrup, Denmark). Plasma FAA was measured by an enzymatic colorimetric assay for the quantitative determination of non-esterified fatty acids (Wako Chemicals GmbH, Neuss, Germany) using the autoanalyzer Cobas C-111 (Roche Diagnostics, Rotkreuz, Switzerland). Plasma glucose was determined applying the enzymatic reference method GLUC assay (Roche Diagnostics GmbH, Mannheim, Germany) on a Cobas c111 system (Roche Diagnostics, Rotkreuz, Switzerland). Plasma β-hydroxybutyrate was measured enzymatically with kit from Randox (Randox Laboratories Ltd., Crumlin, United Kingdom) using the Cobas C111 system (Roche Diagnostics, Rotkreuz, Switzerland).

Throughout the fasting period, all urine was collected in plastic containers, and participants provided 24 h collections of urine each morning. The collected urine was weighed, and its volume density was measured, using a 4 × 1000 μl pipette on a scale. The volume was then calculated, and aliquots were stored at − 70 °C. Total nitrogen was measured according to the Kjeldahl method[48].

## Muscle biopsies and processing

Muscle biopsies were obtained from *m. vastus lateralis* using Bergström needles, both before the fasting period started and after seven days of fasting. The biopsies were taken between 6.00 and 8.00 a.m. The biopsy before the seven days' fasting was taken after an overnight fast. Prior to the biopsy procedure, a local anaesthetic, Xylocaine hydrochloride (10 mg/ml; Astra Zeneca), was applied to the skin, followed by a small incision. The Bergström needle was inserted in the muscle, and suction was used to achieve a vacuum. Muscle biopsies were rapidly rinsed in cold saline, briefly dried on filter paper, immediately frozen in liquid nitrogen, and stored at − 70 °C.

Muscle biopsy samples were freeze-dried for a minimum of 48 h and subsequently dissected free of visible non-muscle tissue. The dissected muscle samples were homogenised in ice-cold lysis buffer (10% glycerol, 20 mM sodium pyrophosphate, 1% NP-40, 2 mM phenylmethylsulfonyl fluoride [PMSF], 150 mM sodium chloride, 50 mM HEPES, 20 mM β-glycerophosphate, 10 mM sodium fluoride, 1 mM EDTA, 1 mM EGTA, 10 mg ml$^{-1}$ aprotinin, 3 mM benzamidine, 10 mg ml$^{-1}$ leupeptin, and 2 mM sodium orthovanadate, pH 7.5) by steel beads and a TissueLyzer II (Qiagen, Hilden, Germany). Tissue homogenates were rotated end-over-end at 4 °C for one hour before part of the homogenates were centrifuged at 18,000 × $g$ for 20 min at 4 °C to obtain tissue lysates (supernatant). Tissue homogenates and lysates were collected, frozen in liquid nitrogen and stored at − 70 °C until further use. For more details about tissue processing and immunoblotting see previous report[49]. Total protein abundance in tissue homogenates and lysates was determined in triplicate by the bicinchoninic acid method (Pierce BCA protein assay kit #23227, Thermo Fisher Scientific, Waltham, MA, USA).

## Immunoblotting analyses and antibodies

For immunoblotting, an aliquot of each muscle lysate and homogenate sample was prepared and boiled in Laemmli sample buffer before being subjected to SDS-PAGE on self-cast Tris-HCl polyacrylamide gels. Proteins were transferred to PVDF membranes by semi-dry Western blotting. Membranes were blocked for 10 min in TBS-T (10 mM Tris-base, 150 mM NaCl and 0.25% Tween 20) containing 2% non-fat milk or 3% bovine serum albumin and subsequently probed with primary and secondary antibodies. Membrane-bound proteins were visualised with chemiluminescence using a digital imaging system (ChemiDoc MP System, BioRad, Hercules, CA, USA) and quantified using Image Lab software (BioRad, Hercules, CA, USA).

Glycogen content in skeletal muscle was measured on 150 μg homogenate after acid hydrolyzation by boiling in 75 μl 2 mM HCl for two hours[50]. Glycosyl units were measured in the supernatant using a Pentra C400 (Horiba Medical, Triolab, Denmark) and given as mmol glycosyl units per kg protein. Glycogen synthase (GS) activity was measured with $^{14}$C-UDP-glucose as substrate and alcohol precipitation of glycogen. The activities were done in 150 μg muscle homogenate and performed in 96-well microtitre plates. Details have been described[51]. Total GS activity was determined in the presence of 8 mM glucose-6-phosphate (G6P). The fractional velocity (FV) of GS was determined in the presence of 0.17 mM glucose-6-phosphate (G6P) and calculated as: 100 × activity in the presence of 0.17 mM G6P / activity at 8 mM G6P.

Antibodies against PDH (E1α)-S293 (site 1), PDH (E1α)-S300 (site 2), PDH(E1α), GS site 1a (corresponding to S698), GS site 2 + 2a

(corresponding to S8 and S10) and GS site 3a + 3b (corresponding to S641 and S645) were produced in collaboration with Prof. Henriette Pilegaard (University of Copenhagen, Denmark) and Prof. D. G. Hardie (University of Dundee, Scotland, UK). Commercial antibodies against Hexokinase II (sc-130358) and AMPKα2 (sc-19131) were from Santa Cruz Biotechnology, while antibodies against citrate synthase (ab96600), representative components (NDUFB8, SDHB, UQCRC2, COXII, ATP5A) of the mitochondrial respiratory complexes I-V (ab110411) and FATP4 (ab200353) were from Abcam. Antibody against PDK4 (PB9773) was from BosterBio, and ACC protein was determined using horseradish peroxidase–conjugated streptavidin (016-030-084) from Jackson ImmunoResearch. Secondary antibodies were from Jackson ImmunoResearch and included goat-anti-rabbit IgG-HRP (#111-035-045), goat-anti-mouse IgG-HRP (#115-035-062), rabbit-anti-goat IgG-HRP (#305-035-003) and rabbit-anti-sheep IgG-HRP (#313-035-003).

## AMPK activity

AMPK activity was measured on consecutive IPs from 200 μg of muscle lysate protein in 96-well PCR plates (first IP AMPK γ3 pulling down the α2β2γ3 complex, second IP AMPK α2 (α2β2γ1) and third IP AMPK α1 (α1β2γ1)). The γ3 and α2 antibodies were custom-made by MRC PPU Reagents and Services, University of Dundee, Scotland and the α1 antibody was custom-made by Genscript, USA. After each overnight incubation at 4 °C with rotation, the IPs were washed once in the IP-buffer, once in 480 mm Hepes (pH 7.0) and 240 mM NaCl, and twice in 240 mM Hepes (pH 7.0) and 120 mM NaCl, leaving 10 μl of agarose after the last wash. The reaction was started by the addition of 30 μl of Reaction-mix containing 80 mM Hepes (pH 7.0), 40 mM NaCl, 833 μM DTT, 200 μM AMP, 100 μM AMARA-peptide (Schafer-N, Denmark), 5 mM MgCl$_2$, 200 μM ATP and 55.5 kBq [γ -$^{33}$P]-ATP (Hartmann Analytic, Germany). The reaction ran for 30 min at 30 °C and was stopped by the addition of 10 μl of 1% phosphoric acid. For detection 20 μl incl. Protein G agarose was spotted onto P81 chromatography filter paper (Saint Vincent's Institute, Medical Research, AUS), which was then washed three times for 15 min in 1% phosphoric acid. The dried filter paper was analysed for activity using a Typhoon FLA 7000 IP scanner (GE Healthcare, Denmark) and by using liquid scintillation (Tri-Carb 2910 TR, Perkin Elmer, Denmark).

**LC-MS measurement of metabolites.** Metabolomics and lipidomics analyses were performed as previously described[52]. In brief, samples were extracted using a methanol-chloroform-water extraction to separate the aqueous and organic fractions[53]. Analytical grade solvents were sourced from Sigma-Aldrich. The aqueous fraction was reconstituted in 10 mM ammonium acetate internal standard solution containing 10 μM proline, valine-D$_8$, leucine-D$_{10}$, lysine-U-$^{13}$C, glutamic acid-$^{13}$C, phenylalanine-D$_5$, succinic acid-D$_3$, and serotonin-D$_4$. Targeted LC-MS/MS was performed with a Thermo Scientific Vanquish UHPLC$^+$ coupled to a TSQ Quantiva mass spectrometer (Thermo Fisher Scientific). Reverse phase chromatography was performed with an ACE Excel 2 C18 PFP (100 A. 150 × 2.1 mm 5 μm) column conditioned at 30 °C using a mobile phase of (A) 0.1% formic acid in water and (B) 0.1% formic acid in acetonitrile. The electrospray ionisation (ESI) source was operated in positive and negative ionisation mode, and nitrogen at 48 mTorr and 420 °C was used for solvent evaporation. Data acquisition and processing were performed using Thermo Xcalibur software (version 2.2; Thermo Scientific), and peak intensity was normalised to appropriate internal standards.

The organic fraction was reconstituted in 1:1 chloroform-methanol containing the following deuterated standards at 2.5 μg/mL: C16-D$_{31}$ Ceramide, 16:0-D$_{31}$-18:1 PA, 16:0-D$_{31}$-18:1 PC, 16:0-D$_{31}$-18:1 PE, 16:0-D$_{31}$-18:1 PG, 16:0-D$_{31}$-18:1 PI, 14:0 PS-D$_{54}$, and 16:0-D$_{31}$ SM (Avanti Polar Lipids); and 18:0-D$_6$ CE, 15:0-D$_{29}$ FA, 17:0-D$_{33}$ FA, 20:0-D$_{39}$ FA, 14:0-D$_{29}$

LPC-D$_{13}$, 45:0-D$_{87}$ TG, 48:0-D$_{83}$ TG, and 54:0-D$_{105}$ TG (CDN Isotopes/ QMX Laboratories). Samples with internal standards were diluted 1:20 into a 2:1:1 isopropyl alcohol-acetonitrile-water solution with 10 mM ammonium formate added to solvents for positive ionisation and 10 mM ammonium acetate added to solvents for negative ionisation. Chromatography was performed with a 75 μM x 100 mm C18 packed-tip column (Thermo Scientific) conditioned at 55 °C using a mobile phase of (A) 60:40 acetonitrile-water and (B) 10:90 acetonitrile-isopropanol. Open-profile MS was performed using an Orbitrap Velos Elite Mass Spectrometer (Thermo Scientific) in positive and negative ionisation mode. Peak integration was performed with XCMS software in R studio[54]. Annotation was performed using accurate mass and reference to the human metabolome database (https://hmdb.ca/) and LipidMaps (https://lipidmaps.org/), excluding analytes that could not be annotated from further analysis. Peak intensity was normalised to the internal standard of the corresponding lipid species.

## Statistics

Pre- and post-values are reported as means ± SEM. A paired two-sided Student's t-test was used to compare pre- and post-values for parameters measured twice, such as $\dot{V}O_{2peak}$, maximal isometric strength, and muscle glycogen. A two-sample two-sided Student's $t$ test was employed to compare data between males and females. Exercise tests involving multiple measurements were analyzed using repeated measures ANOVA, with Fisher's LSD post-hoc tests applied where appropriate.

Metabolomics data are from 11 individuals and were analysed using repeated measures two-way ANOVA with Tukey's multiple comparisons test as the post hoc test. Metabolite abundance was normalised to internal standards ("relative abundance"). Metabolites with > 20% missing values (below the detection limit) were removed from the dataset. All other missing values were imputed with a 20% minimum value. Linear regression analyses were used to establish the relationship between baseline values and responses. $P$-values are reported in Tables and Figures. In figures with metabolomics data, differences are marked with asterisks; [*] $p < 0.05$, [**] $p < 0.01$, [***] $p < 0.001$, [****], $p < 0.0001$.

## Reporting summary

Further information on research design is available in the Nature Portfolio Reporting Summary linked to this article.

## Data availability

The data are not publicly available. We do have not permission to make the data publicly available because the data may contain information that could compromise participants' privacy and consent. Data to support the findings of this study are available on request to Jørgen Jensen, Department of Physical Performance, Norwegian School of Sport Sciences, Post box 4014 Ullevål Stadion, 0806 Oslo, Norway, E-mail: jorgen.jensen@nih.no. The response will be provided within two weeks. Data sharing requires approval of the Ethics Committee and Norwegian Centre for Research Data (Sikt), followed by institutional approvals of data sharing agreements. Source data are provided in this paper.

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

## Acknowledgements

We acknowledge all the participants who volunteered for the project. We thank Marte Valde for blood sampling and great organisational work in the laboratory during the clinical trial. We thank Prof. Henriette Pilegaard for providing PDK4 antibodies and Dr. Anders Gudiksen for advice on the analyses. The clinical part was funded by the Norwegian School of Sport Sciences. Novo Nordisk Fonden #082659 to J.F.P.W. UK Medical Research Council (MR/P011705/2; UKDRI-5002; MAPUK) to J.L.G.

## Author contributions

Conceptualisation: K.J.K., E.T.F.N., S.B., E.I.J., A.J.K., S.O.R., J.F.P.W. and JJ. Data curation: K.J.K., E.T.F.N., S.B., A.M.M., P.B.J., Ø.S., J.B.B., J.H., R.K., A.J.K. and JJ. Formal analysis: K.J.K., E.T.F.N., S.B., A.M.M., P.B.J., Ø.S., J.B.B., R.K., J.L.G. and J.J. Funding acquisition: J.F.P.W. and JJ. Project Administration: J.J. Writing—original draft: K.J.K. and JJ. Writing—review and editing: K.J.K., E.T.F.N., S.B., A.M.M., P.B.J., Ø.S., E.I.J., J.B.B., K.H., J.H., B.S.S., R.K., J.L.G., A.J.K., S.O.R., J.F.P.W. and J.J.

## Funding

## Competing interests

J.F.P.W. has ongoing collaborations with Pfizer Inc. and Novo Nordisk A/S unrelated to this study. J.F.P.W. holds shares in Pfizer Inc. and Novo Nordisk A/S. During 2024 J.F.P.W. was a consultant at Pfizer Inc. K.J.K. was appointed at Novo Nordisk A/S June 2024. The remaining authors declare no competing interests.
