## [Peer Review file · Nature Communications]

Effects of Seven Days' Fasting on Physical Performance and Metabolic Adaptation during Exercise in Humans

Corresponding Author: Professor Jørgen Jensen

Version 1:

Reviewer comments:

Reviewer #1

(Remarks to the Author)

The authors have comprehensively responded to the concerns of this reviewer.

Reviewer #4

(Remarks to the Author)

Reviewer #5

(Remarks to the Author)

The authors have provided an extensive and well-argued response to the reviewers. I have no further comments, except the placing of the data in the context of evolution. You have no data on evolution, and it is enough to refer to recent history to assess the significance of physiological function, such as muscle strength and endurance, when we think of the severe famines in the 70's of the last century in Sharan countries and Bangla Desh. I would refrain from the evolutionary inference. Some comments in the annotated manuscript related to this comment.

REVIEWER COMMENTS

Reviewer #1 Comments:

Kolnes and colleagues examined the joint effects of water-only fasting and exercise on physical performance and metabolic changes in a very small sample of apparently healthy younger adults. The results are remarkable for the finding of preserved strength but a decline in endurance due to fasting. Various unexpected changes and lack of changes in metabolic biomarkers during exercise while fasting are of note and provide impetus for further studies. The study was generally performed well and the paper provides a wealth of data.

Answer: Many thanks for your positive evaluation.

Major comments:

1. On line 76, 2 individuals are reported to have withdrawn from the study due to noncompliance with the intervention. Was there a reason given by those 2 people about why they stopped fasting?

Answer: According to Norwegian law, participants can withdraw without any explanation, and we did not pursue the question. However, one of the participants withdraw after only one day of fasting because she prepared for an important exam. This participant had a friend who completed the fasting period, and we had some contact with the "dropout" afterward. The other participant withdrew after 2-3 days and gave as explanation that her working schedule had been changed, which made it impossible for her to complete the study according to requirements. We did not ask for additional information.

2. On line 185, given the multiple testing issues in this study that is evaluating multiple hypotheses and does not apply any correction for multiple comparisons, saying the following is somewhat of a stretch: "ACC tended to be upregulated ($p=0.09$).". A p-value above 0.05 for a secondary/exploratory outcome should be considered not significant. In conjunction with this point, it would also be good to mention in the Methods that p-values of $p \leq 0.05$ for secondary hypothesis tests are considered either confirmatory of prior published data on fasting and/or exercise, or are exploratory results requiring prospective confirmation in future studies (or to correct for multiple comparisons, such as with a Bonferroni correction or other approach). It would be worthwhile to mention that Tukey's test, for example, corrects for multiple comparisons since the T-test that is used in most of the analyses does not do so. The primary hypothesis testing for the evaluation of maximal leg muscle strength (including for tests of correlated outcomes of muscle strength) is acceptable to be tested at $p \leq 0.05$ as has been done.

Answer: Many thanks for your comment and we completely agree. Indeed, the Western blot are secondary outcomes, and we now report ACC as unchanged in the revised version. Said that, we were surprised that the proteins involved in fat oxidation was unchanged. However, we (David E James, Sydney) have done proteomics on muscle biopsies from the other cohort (Pietzner et al., Nature Metabolism) and expression of only 4 proteins changed significantly after the fasting period after Bonferroni correction, and none of the proteins were metabolic enzymes or contractile proteins (MHC or actin). The proteomic data also showed that PDK4 was upregulated, but not significantly after multiple comparisons. These data support our findings that expressions of

metabolic enzymes were not changed significantly by 7 days fasting. We agree that our data do not support any significant change in ACC and have changed accordingly.

3. On line 211, a citation to a recent paper by Pietzner et al is made. That paper by Pietzner evaluated a very similar population of 12 individuals and the design followed similar 7-day fasting interventional procedures with similar inclusion/exclusion criteria. Was this study performed in that same population, or did it just use the same design? If it was the same population of participants, this should be mentioned and an explanation given as to why this study had one more participant than the Pietzner study.

Answer: We apologise for the confusion. The study by Pietzner et al. is from a completely different cohort with 12 individuals. There were 13 individuals included in the cohort used for the current paper. Indeed, the design was rather similar, but we did not include exercise testing during fasting period in the cohort reported by Pietzner et al. In the cohort by Pietzner, we followed weight gain three days after returning to their normal eating habits. We apologise the confusion and have clarified (see also response to Editor).

4. The Limitations section should note on line 311 that, "..., the sample size was very small." Also, the limitations include and that section should state that causal inference is limited by the lack of a parallel control group and no randomization of participants to exercise or fasting interventions.

Answer: Many thanks for your comment. We have now included that we lack a parallel control group and no randomization of participants to exercise or fasting interventions as a limitation.

5. No discussion of the method of ensuring adherence to the water-only 7-day fasting regimen (i.e., no caloric intake) is provided. While some biomarkers are available and discussed in the paper that could be used in that assessment, these are not mentioned regarding whether they agree with the idea that adherence to no caloric intake was achieved during the 7 day fasting period. Further, were any questions asked of participants during daily assessments across the 7 days of fasting to ensure that subjects were fasting?

Answer: Many thanks for the suggestion and question. Participants were equipped with a continuous glucose monitor, which allowed to measure glucose thorough the whole fasting period. However, we acknowledge that protein/fat ingestion would be hard to catch with the measurements in this cohort. However, the daily blood samples measuring ketones and FFA (increased FFA and β -hydroxybutyrate) indicate that they adhered to the fasting regime. We have now commented that in discussion. Many thanks.

We were in daily contact with the participants, and they completed a questionnaire daily (Translated version attached). Indeed, we did not have a direct question about whether they had followed the fasting regime; we asked about their desire to interrupt and motivation to continue. The participants scored low at the desire to interrupt (0.8 cm – VAS 10 cm) and high for the motivation to continue (8.9 cm). In our daily contact, which included measurement of body weight, we repeatedly explained their valuable contribution by water-only fasting for our research. The gradual weight reduction also supports the participants adhered to fasting.

Additionally, the participants were not paid for participation and there was no economic benefit from cheating. Participants were informed that they could withdraw at any time without any explanation.

6. In Figure 2 on panels D, E, and F, just one threshold of p-value should be used along with a less-than sign (" $<$ "). Are the p-values $p=0.00001$, $p=0.0004$, and $p=0.000001$ or is the result for each analysis truly less than the stated values? Using one threshold to assign a " $<$ " sign to is sufficient in the paper, such as $p<0.0001$ given the many tests performed and reported in the paper. Figure 3 also uses different thresholds for $p<0.0001$ and $p<0.00000001$ in different panels. Figure 4 uses $p<0.001$. Using $p<0.0001$ should be sufficient in all of these Figures, and in other Figures and in all of the Tables and text, to show that a statistical test is highly significant. Usually, one would limit it to $p<0.001$, but given all of the statistical tests performed here, $p<0.0001$ is perhaps more warranted. All of the figures and tables should be standardized so that p-values above or equal to 0.0001 are displayed as $p=$ the actual p-value. Anything below $p=0.0001$ would be displayed as $p<0.0001$. The approach in the supplementary figures of having asterisks for ranges of p-values is acceptable (although to be clear it appears that the grouping scheme assumes, for example, that " $*p<0.05$ " actually means " $p<0.05$ and $p\geq 0.01$ " while " $***p<0.001$ " means " $p<0.001$ and $p\geq 0.0001$ " and so forth.

Answer: Many thanks for your comments and suggestions. We apologise the confusion. We have revised the figures and now consistently report significant differences as " $<$ ". In the previous version we reported some of the differences as e.g. $p=0.0004$, but these numbers were always rounded up; therefore, these values were always " $<$ ". According to your suggestion, we selected to report data with two, three, or four decimals and restricted to use " $0.05, <0.01, <0.001, 0.0001$ " to designate significant differences. Regarding p-values for comparisons that are shown in tables, we still report these values with three decimals (Table 2). However, we can change or remove these p-values on request.

Minor comments:

1. On line 334, the phrase, "..., but the ability to withhold intense aerobic work..." doesn't make sense. Should it actually say, "..., but the ability to sustain intense aerobic work...?"

Answer: Many thanks. We agree and have revised. Your interpretation of the sentence was correct.

2. A variety of typographical errors are found throughout the paper, thus the text should be checked for spelling, punctuation, and other errors. For example, The legend for Figure 1 on lines 588-589 highlight this need where the word "deigns" is used but likely it was meant to say, "design" as deigns means something completely different and not related to the topic. Also in that sentence, it says "intestinal" and likely should be "interstitial" and later it says, "fasting.and" where the period is misplaced.

Answer: Many thanks for your corrections. We have now used professional language edition service. We hope all typographical errors are corrected.

We thank the reviewer for a careful review and most useful comments for revision. We find that your comments contributed to a better manuscript.

Reviewer #2 Comments:

The authors performed a very unique study that involves 7 days of fasting of male and female human subjects and tracked body weight, body composition and several physiological and metabolic

measures to gain insights into the changes in performance and metabolism. Some of this work was published in Pietzner et al, Nature Metabolism 2024).

Answer: Many thanks for your positive evaluation. In fact, the present data are from a different cohort (see also answer to Editor). We apologise the confusion. We have now included a sentence in the revised manuscript specifying that the data in the present manuscript is from another cohort (see also answers to Editor and Reviewer #1).

The subjects were young healthy adults (~30yrs) with good starting fitness (~48 ml/kg/min $\dot{V}O_{2max}$). As expected the subjects lost body weight with contributions from lean and fat mass. While strength measures did not change, there was a significant reduction in $\dot{V}O_{2max}$. Metabolic measures point to a significant increase in fat metabolism (with some increases in plasma leucine, valine and isoleucine) with a decline in CHO metabolism. The authors did obtain muscle biopsies and probed some signaling pathways associated with metabolism and found changes in PDH and PDK4 as the most significant changes consistent with the metabolic transitions. However, there are some areas not addressed;

1. The authors did not discuss potential changes in total body water. Was this measured? Is water part of the lean body mass loss? this needs to be discussed as there would likely be some water changes with fasting.

Answer: Unfortunately, we did not measure total body water. Body water probably makes a substantial contribution the weight loss according to previous studies. Bloom et al. found that after 4-7 days of fasting (avg. 5.5 days) reduced plasma volume by 498 ml (Bloom et al. 1966, Metabolism Vol 15:409-413). We have made additional calculations of water that will bind to glycogen. Assuming that 35% of the body weight is muscle and a mean reduction of muscle glycogen of 200 mmol/kg dry weight (as we observed), it will cause a reduction in muscle glycogen of ~160 g. Assuming each g glycogen binds 3 g water, the muscle glycogen will bind ~480 g water. Assuming a reduction of 80 g liver glycogen, additional 240 g water will be released. Reduction of body glycogen may therefore account for 240 g and release 720 g water. The reduction of 524 g protein (according to urine nitrogen excretion) will reflect 2.6 kg cell mass and ~1.8 kg water assuming 70% of cells is water. In addition, some water will be lost via faeces as the intestines become emptied during the fasting period. In the present study weight loss was 5.8 kg. The weight loss in different body compositions were 1) 1.4 kg fat, 2) 2.6 kg cell mass (calculated from nitrogen excretion) and 3) 0.96 kg glycogen + water. Indeed, it would have been interesting to have data on total body water, which would have allowed a more thorough interpretation of the data.

2. The authors report continuous glucose monitoring that shows a decline through day 3. They did not probe any potential changes in the daily pattern of glucose from these records. Since the subjects were not eating, it would be important to know if the levels of glucose were absolutely flat throughout the day or whether there were any basal oscillations in circulating glucose?

Answer: The participants were equipped with Dexcom and we have glucose values every 5 min during the 7 days of fasting. However, the study was not intended to investigate the glucose pattern during fasting. The CGM was equipped for safety reasons and to ensure that participants were fasting. We have looked at the data again. Indeed, the glucose was not flat during the fasting period. During the exercise test, we report an increase in plasma glucose from 3.8 ± 0.2 mM to 5.6 ± 0.2 mM after the $\dot{V}O_{2max}$ test. The participants had no exercise restrictions, and CGM values showed variation

during the fasting period. We have calculated CV for CGM on day 5 which varied between 5% and 18% for the participants (mean 12%; STDEV 4%). We have not analysed the CGM data further.

3. With the significant declines in lean mass it was surprising to see no change in knee extensor strength. The authors did obtain muscle biopsy material yet they did not include any analysis of the amount to total protein/mg muscle mass. If strength was maintained it would be hypothesized that this ratio would be maintained. For this analysis, it is important to solubilize all the proteins with a strong buffer like urea, or higher salt - which is different compared to the buffer used to assess signaling molecules as outlined in the manuscript.

Protein analysis of myosin heavy chain amount to protein would also be an informative measure for these performance assays. Not myosin isoform but just how much myosin heavy chain is there relative to the total protein? If the subjects are losing protein, are the contractile proteins being spared?

Answer: This was an expert question and difficult to answer. We also expected a decrease in muscle strength and was surprised. We have looked at the data on protein analysis again. In the present study, we have only extracted ~580 mg protein/g dry weight, confirming your expectation that not all protein was solubilised. However, there was no difference in recovery between pre and post biopsies. We agree that connective tissue may not have been dissolved adequately with the buffer used as it would have been expected to extract ~800 mg protein/g dry weight. The muscle biopsy material has been used and we cannot make a new extraction. It would be possible to run gels and silver-stained for MHC from the samples we have. However, with a recovery of 58% such data may not provide any valuable information. In the present study we find that 524 g protein is degraded (according to nitrogen excretion). Assuming 35% of a person is muscles, that will correspond to ~5 kg pure protein. We believe it will be impossible to address degradation of specific proteins with the material we have left. We completely agree in your concern, and we have modified the paragraph in Discussion where we discuss protein degradation.

4. Minor point; please note the general time day in which the testing/biopsy collection was performed. Several metabolic measures exhibit time of day variation (resting metabolic rate, RER - as well as blood and muscle measures).

Answer: Many thanks. We have included information about the time the different tests were performed.

We thank the reviewer for most insightful comments.

Reviewer #3 Comments:

The manuscript entitled: 'Effects of Seven Days Fasting on Physical Performance and Metabolic Adaptation during Exercise in Young and Healthy Individuals' demonstrated preservation of muscle strength, while a reduction in aerobic capacity in response to prolonged fasting.

The study is well designed and for a 'diet intervention' study, is well controlled, especially given their participants to be 'free living'. The findings are interesting, but probably not surprising given the outcomes measured. I am struggling to agree with the manuscript narrative and direct links between

findings and human's evolutionary requirement to gather food during times of starvation. The conclusions of the paper regarding 'maintenance of muscle strength and oxidative enzymes during fasting may have enabled humans seeking alternative food sources during periods of severe shortage', while I don't disagree that this might have been the case, I'm not sure that a direct link can be made based on the study findings. Firstly, muscle strength was determined by three isolated knee extensions (at various angles). For those looking for food, more than 3 max contractions would be required. Similarly, with a VO₂max test on a bike. For translatory purposes, why not utilise a submaximal time to exhaustion protocol on a treadmill, presuming our ancestors would utilise low intensity movement to conserve energy.

Answer: Many thanks for your positive comments about the study design. We agree that that the link between the findings in the present study and human evolution is not a proof for evolutionary biology. Furthermore, despite physical capacity has been crucial for survival, we also acknowledge that the physical tests in the laboratory setting are far from the "free-living" conditions in an evolutionary perspective.

We discussed several protocols for the physical tests when designing the study. We decided that the physical tests should be performed in the morning of day 6. The effect of fasting on muscle strength in the leg had never been investigated, and this test had highest priority. We discussed several options and decided to test dynamic isokinetic strength at three velocities (60, 120 and 180 degree/s) and isometric strength at 60 degrees angle in knee joint. We consider the strength tests used in the present study good and by far the most comprehensive investigation of muscle strength during fasting. We also wanted to test the capacity for fat oxidation and maximal oxygen uptake. Indeed, other or additional tests could have been selected, but we were afraid that more extensive testing would reduce quality of the tests.

We agree that a treadmill test would have been better for translatory purpose, and such test would have been more relevant in evolution perspectives. We completely agree that low intensity physical activity would have been used during evolution, and the intensity would have been adapted to physical capacity. Unfortunately, submaximal tests to exhaustion on a treadmill are very difficult to conduct and get reliable data. Furthermore, the reduction in body weight would reduce energy requirement during treadmill exercise. Of note, Consolazio et al. (1967; Ref #5 in the manuscript) reported a 19% reduction in maximal oxygen uptake as L/min after 10 days fasting, but no significant reduction in performance on a treadmill walk test (Balke-test; gradually increase in slope of the treadmill until exhaustion). We decided to use a bike test to obtain absolute values in work capacity.

In the present study we aimed to test the capacity for high intensity exercise (VO_{2max}) and selected bike test to get data on power output. We consider the findings that capacity for high intensity exercise and carbohydrate oxidation were reduced significant since we provide a likely mechanism (increased PDK4 expression). Indeed, the tests you suggested would have provided interesting data, but we selected other tests, unfortunately.

We think it is unquestionable that physical capacity was important for survival, and this concept was the idea behind the study. We have revised and modified the manuscript and hope it will meet your expectation. However, we argue that the collected data during fasting are relevant in evolutionary perspectives.

The second conclusion 'Fasting seems to require a trade-off with preservation of strength being prioritised over maintenance of endurance capacity', I'm not sure whether I entirely agree and think the results are more of reflection of utilisation of different energy systems within the tests

themselves, available energy sources and/or capacity to use those. For example, the isokinetic/isometric strength tests are utilising some energy systems (PCr/anaerobic) as well as motor unit firing (which is important), whereas the VO₂max test is heavily relying on the aerobic, but more anaerobic systems at the end with increasing intensity. It was clear that RER was lower after the fasting suggesting a lower ability to use carbs (as evident by the protein markers). In addition, little change in mitochondrial enzymes (though hard to determine given low sample size) may also reflect maintenance of endurance capacity per se.

Answer: We have changed the second conclusion to “However, the capacity for carbohydrate oxidation and high intensity endurance exercise decreased” (Abstract) and to “However, whereas muscle strength was preserved there was a 10-15% decrease in high intensity endurance capacity” as the last sentence in Introduction. We agree that we do not have any data to support a direct link between the reduction of aerobic capacity and maintenance of maximal strength.

We consider it a key finding that the carbohydrate oxidation and peak oxygen uptake were reduced so dramatically. Importantly, we report a likely mechanism for that, and we are the first to study muscle biopsies beyond three days of fasting, and the 13-fold increase in PDK4 is novel. The efficient reduction in carbohydrate oxidation was documented by the low RER despite the high blood lactate concentration. We think this reduction of carbohydrate oxidation during exercise is an important mechanism in an evolution perspective.

The isokinetic tests at three velocities and isometric strength were included to test maximal strength and we agree that the challenge on anaerobic energy systems was low. The idea was to test maximal strength. Therefore, participants were allowed 1 min rest between the isokinetic tests at different velocities, and between the isometric tests for recovery of PCr. Indeed, motor unit firing would have been interesting to investigate, but we did not have this expertise or the ability to include such testing.

We completely agree in your conclusion, that the small changes in mitochondrial enzymes may reflect maintenance capacity per se (see also answer to referee #1 point 2). The effect of fasting on physical capacity needs future investigations, but we still think the present paper report important data.

There does appear to be quite heterogeneity in some of the outcomes reported and the non-significance could be a reflection of low sample size. Was effect size or power calculations undertaken to determine strength of the statistical analysis?

Answer: Many thanks for your comments. We find that most of the outcomes are very consistent, especially the clinical ones; VO₂max, RER, reduction in weight and body composition. We also find the data on substrate oxidation during exercise are consistent. Additionally, all participants showed a decrease in glycogen (Indeed resting glycogen varied, but that is normal) and PDK4 increased in all participants. Of note, the initial body weight of the participants varied between 53 and 114 kg which causes heterogeneity in some parameters, but urinary nitrogen excretion and reduction in body fat mass correlated with initial body weight (and LBM).

Post hoc power calculations of statistical analysis are not recommended, and we prefer not to do so. We find that many of the key findings in the study are quite clear. Indeed, some data e.g. the Western blots show variation and statistical power is not high despite we have included 12 participants in these analyses, which is not a low number in studies with muscle biopsies.

More details regarding muscle strength testing is needed in the methods. What machine was used? Where were the hands/arms placed as this can influence results between participant and tests.

Answer: We apologise for not including information about strength tests and many thanks for pointing this out. We have included information about the strength tests in the revised version.

Isokinetic and isometric strength were tested with a Humac dynamometer (Humac NORM, Computer Sports Medicine Incorporated (CSMi), Stoughton, MA, USA). The participants were seated in an upright position and the chair was adjusted to have the rotational axis of the knee in the dynamometer's axis of rotation. The lever arm was secured just proximal to the ankle joint with a trap. The backrest (hip angle) was set at an angle of 85° and participants were secured with four points trap according to recommendations and kept their hands in the lap. Chair adjustments were noted and used in familiarisation, pre- and post-tests.

Given the different loss of LBM to fat mass and thus overall body weight, why not present changes in VO₂max relative to LBM? Could power output on bike also be presented relatively.

Answer: Many thanks for your comment. In exercise physiology, VO₂max is mostly reported in absolute values and relative to body weight, and we have done that in all our previous studies. We have included a figure with VO₂max relative to LBM shown below. We can also include the figure with VO₂max related to LBM in the manuscript on request.

We have included a new figure in the revised manuscript showing maximal power output at the bike test relative to body weight as suggested (Figure 2G in the revised manuscript).

Figure 1. Effect of fasting on maximal oxygen uptake as ml/kg LBM/min.

Minor:

Line 44: Add context. Duration of fasting, as an overnight 'fast' should not decrease lean mass.

Answer: Many thanks for the comment. We have now included information that body composition was investigated after six days of fasting in the study we refer to.

Line 118 – carbohydrate needs a e

Answer: Many thanks. We have corrected.

Line 196: reference after in the fed state

Answer: Many thanks. We have included a reference to the statement.

Line 202: It is possible that this represent the primary...should it be 'represents'

Answer: Many thanks. We have revised.

Line 300: I'm not sure Biodex and VO₂max testing is 'state-of-the-art methodologies'

Answer: Many thanks for your comment. Maximal strength is not an easy parameter to test. We have previously used 1-RM (1 repetition maximum) in intervention studies, but we did not find that test suitable for the present study. Neither did we find hand grip strength, as used in previous fasting studies, was suitable. Therefore, we selected to test isokinetic strength in Humac dynamometer. Although isokinetic strength and measurement of VO_{2max} by indirect calorimetry have been used for many years, these tests are still preferred, and we think it is fair to call them 'state-of-the-art methodologies'.

Did participants reach the criteria for VO₂max? If not, then it should be presented as VO₂ peak

Answer: Many thanks for your comment. We have changed to VO_{2peak}. We normally use as criteria for successful VO_{2max} 1) levelling off, 2) RER > 1.1, 3) blood lactate > 7 mM, and 4) voluntary exhaustion (18 or above on the Borg scale). Fulfilment of at least 3 of the criteria are required for successful measurement. In the present study, RER did not reach 1 during fasting. Lactate and voluntary exhaustion criteria were achieved. However, levelling off was also difficult to obtain in all participants. It is our experience that it is difficult to obtain "levelling off" at VO_{2max} on a bike for participants who do not perform bike training.

Did anyone exercise during the fasting period and was this monitored?

Answer: Participants had no restriction regarding exercise or daily activities. The participants wore Actiheart for continuous measurements of heart rate and movement the week of fasting. In addition, the participants wore an Actiheart a week before the fasting period. In the 13 participants included in the present study, energy expenditure did not differ significantly between the week prior to and during the week of fasting. Our plan is to publish these data in a separate paper together with data from the other cohort (Pietzner et al., 2024 Nature Metabolism) to increase statistical power.

Why didn't you utilise a fatiguing protocol on the biodex/cybex machine- this could further/better support your claim regarding differences in strength and endurance capacity?

Answer: Many thanks for your comment. Indeed, a fatiguing protocol in the Humac dynamometer would have been of interest and provided further support for our conclusions. However, as explained above, we discussed various protocols and decided to test maximal isokinetic strength (three velocities), isokinetic strength, fat oxidation and maximal oxygen uptake. We were afraid that a too comprehensive test regime would deteriorate the test results. In total these tests lasted around one hour, and the tests were quite demanding. In the future, it will be important to do more investigations of the effect of fasting on physical capacity to get a more comprehensive understanding of the physiological adaptations.

Still, we find that the present study presents important data and provide impetus for further studies.

We thank the reviewer for most careful assessment of the manuscript and most valuable comments.
We hope our answers and revision are satisfactory.

Dear Reviewer #5

Many thanks for most thoughtful suggestions and positive evaluation. We completely agree and have changed “evolution” to “history” as suggested.

We have revised the manuscript according to the comments from reviewer. The revised manuscript is in accordance with all the suggestions included in the manuscript.

Yours sincerely,

Jorgen

22 Abstract

23 Humans have, throughout ~~evolution~~ history (*while evolving they were not human!*), faced periods of
24 starvation necessitating increased physical effort to gather food. To explore adaptations in muscle
25 function, 13 participants fasted for seven days. They lost 4.6 ± 0.3 kg lean and 1.4 ± 0.1 kg fat mass.
26 Maximal isometric and isokinetic strength remained unchanged, while peak oxygen uptake decreased
27 by 13%. Muscle glycogen was halved, while expression of electron transport chain proteins was
28 unchanged. PDK4 expression increased 13-fold, accompanied by inhibitory PDH phosphorylation,
29 reduced carbohydrate oxidation and decreased exercise endurance capacity. Fasting had no impact on
30 AMPK activity, challenging its proposed role in muscle protein degradation. The maintenance of muscle
31 strength and oxidative enzymes during fasting may have enabled humans to seek alternative food
32 sources during periods of severe shortage (*well, this could be, but is yet the next sentence says that*
33 *essentially it did not matter, and in fact was associated with a reduced rather than increased endurance,*
34 *suggesting that the conclusion in this sentence is not supported*). However, carbohydrate oxidation and
35 high-intensity endurance capacity were reduced.

36

37 Total word count: 11020

38 Abstract: 959/118 words

39 Word count main text: 4191/5000

40 Max 8 figures/tables – but additionally 10 figures.

41 References 54 – should be 50.

42

43 Introduction

44 Throughout ~~history (think only about the recent (1970's famines) evolution,~~ humans have faced periods
45 of nutritional shortage, necessitating increased physical effort to gather new sources of food. Humans
46 are well-adapted to tolerate periods without food and most individuals have sufficient fat stores to
47 survive several weeks ¹, but six days' fasting decreases lean mass substantially ², which may impair
48 physical capability. ~~From an evolutionary standpoint, preservation of muscular strength and aerobic
49 capacity likely favoured survival in periods with limited food. (Does not follow logically from previous
50 sentence where it is said that fasting decreases lean mass, hence muscle, nor the following sentence,
51 where again it is mentioned there is protein loss. For the argument it is not needed, so just remove the
52 sentence).~~ During fasting, degradation of protein is the main source of amino acids for gluconeogenesis
53 ³. Whether this protein degradation includes contractile proteins is uncertain, and the impact of fasting
54 on muscle strength has not been thoroughly studied. Grip strength is well-preserved during the first
55 two weeks of fasting, thereafter declining ^{4,5}, while strength in the legs remains unexplored. Endurance
56 capacity declines noticeably after just 24 to 72 hours of fasting ^{6,7}, even though the expression of
57 mitochondrial oxidative enzymes in skeletal muscle has been reported to be unaltered after three days
58 of fasting ^{8,9}.

59 Glycogen stores are limited, and liver glycogen is completely depleted after 24-36 hours without food
60 ^{10,11}. In contrast, muscle glycogen, which is the main substrate during exercise of moderate and high
61 intensity, decreases only by 20-30% after three days of fasting ^{8,12,13}. Muscle glycogen is also the main
62 substrate for anaerobic energy production ¹⁴, required for high-intensity "fight or flight" activities.
63 Knowledge about muscle glycogen regulation during fasting is therefore essential to understand
64 humans' capacity for physical effort during periods of starvation. Additionally, fasting elicits some well-
65 established changes in plasma metabolites (decreased plasma glucose and increased plasma fatty acids
66 (FA) and ketones), which have been studied for up to 40 days of fasting under resting conditions ^{1,15}. A
67 recent study found that the plasma concentration of unsaturated FAs increased much more compared
68 to saturated FAs after ten days of fasting ¹⁶. The corresponding metabolic responses during exercise
69 after prolonged fasting remain unknown.

70 The aim of the present study was to address several unexplored aspects of prolonged fasting and its
71 effects on physical performance and skeletal muscle adaptations. We report preservation of maximal
72 strength in leg muscle, despite a significant loss of lean mass. Further, there was a marked decline in
73 peak oxygen consumption after six days of fasting. We identified a 13-fold increase in pyruvate
74 dehydrogenase kinase 4 (PDK4) expression, increased PDH phosphorylation and compromised
75 carbohydrate oxidation during aerobic exercise, despite unchanged expression of oxidative enzymes
76 and preservation of 50% of muscle glycogen content. Therefore, humans maintain their capacity for
77 physical abilities well during periods of severe food shortage. However, whereas muscle strength was

296 oxidize ketones under hyperketonemia ^{37,41}. From the present cohort, we have previously reported a
297 gradual increase in resting β -hydroxybutyrate (BHB) ⁴², where plasma BHB was almost absent after
298 fasting overnight, but increased to 4 mM after prolonged fasting. In the present study, there was a 28%
299 decrease in BHB during exercise in the fasting state. This finding is in line with earlier reports ^{37,43}
300 supporting the proposition that BHB serves as an energy substrate in muscle. Importantly, ketones
301 inhibit lipolysis in adipose tissue ⁴⁴, and a decline in plasma ketones might be necessary to achieve a
302 high rate of lipolysis in adipose tissue during exercise after prolonged fasting.

303 **Strength and limitations**

304 Dynamic muscle strength in leg muscles after prolonged fasting has never been investigated before and
305 we expanded the duration for investigation of glycogen, expression of metabolic enzymes and AMPK
306 activation in skeletal muscle from three to seven days of fasting. It is a strength that we included healthy
307 participants with training experience and used state-of-the-art methodologies for physiological tests
308 after familiarization tests. Furthermore, we tested both strength and endurance in the same cohort to
309 ensure that the study could evaluate a holistic perspective of physical capacity. It is a strength that
310 changes, both in clinical and metabolic variables, were consistent in all participants, making the data
311 reliable. We consider it a strength that the study was performed in free-living conditions rather than on
312 hospitalized participants, to imitate real-world scenarios. Moreover, we have demonstrated the
313 feasibility of conducting weeklong fasting studies under free-living conditions with CGM, collection of
314 urine, daily blood sampling, weighing and questionnaires. The questionnaires confirmed that the
315 volunteers (unpaid) were motivated to continue, and the increase in β -hydroxybutyrate and FFA in
316 combination with the decline in weight and blood glucose show that the participants followed the
317 fasting regime. There were only two drop-outs early in the fasting period and no serious side effects. It
318 is also a strength that we investigated both sexes.

319 A limitation of the present study is the lack of a randomized control group, which limits casual inference.
320 Moreover, muscle biopsies were not collected before and after exercise, preventing us from isolating
321 the effect of prolonged fasting and exercise on glycogen metabolism and enzymatic activation of
322 relevant proteins. Although 13 participants completed the study, the sample size was relatively small.
323 All participants were healthy, young adults, limiting the generalizability of the results. Furthermore, all
324 tests were done at the same time of day, restricting us from investigating the potential effect of
325 circadian rhythms during fasting. Despite participants doing the study in free-living conditions, today's
326 free-living conditions are far from free-living conditions ~~from an evolutionary perspective in the past~~.
327 Future research with more diversified cohorts is necessary to substantiate these findings, particularly
328 concerning the potential therapeutic effects on various diseases. Finally, the endurance testing protocol
329 could have been more wide-ranging to characterize the metabolic response during exercise more